# ON THE TRADE-OFF BETWEEN EXPRESSIVITY AND PRIVACY IN GRAPH REPRESENTATION LEARNING

**Patrick Indri**[1,*]**, Tamara Drucks**[1,2,*]**& Thomas Gärtner**[1]
[1]Research Unit Machine Learning, TU Wien, Vienna, Austria
[2]Faculty of Computer Science, University of Vienna, Vienna, Austria
{patrick.indri,tamara.drucks,thomas.gaertner}@tuwien.ac.at

## ABSTRACT

We investigate the trade-off between expressive power and privacy guarantees in graph representation learning. Privacy-preserving machine learning faces growing regulatory demands that pose a fundamental challenge: safeguarding sensitive data while maintaining expressive power. To address this challenge, we leverage homomorphism density vectors to obtain graph embeddings that are private and expressive. Homomorphism densities are provably highly discriminative and offer a powerful tool for distinguishing non-isomorphic graphs. By adding noise calibrated to each density's sensitivity, we ensure that the resulting embeddings satisfy formal differential privacy guarantees. Our theoretical construction preserves expressivity in expectation, as each private embedding remains unbiased with respect to the true homomorphism densities. We demonstrate the usefulness of our embeddings through experiments on molecular and social network datasets.

## 1 INTRODUCTION

We study the interplay between expressivity and privacy for learning graph representations and show how to obtain expressive *and* private representations. Our investigation addresses the need for privacy-preserving machine learning and our formal guarantees align with the increasing regulatory pressure in this direction (European Parliament and Council of the European Union, 2016; 2024; National Institute of Standards and Technology, 2023). In graph representation learning, expressivity analysis studies the ability of algorithms to distinguish pairs of non-isomorphic graphs. Private algorithms, on the other hand, generally ensure that similar graphs yield similar outputs. Consider for instance graphs $G_1$ and $G_2$ in Figure 1 that differ by exactly one edge. As the two graphs are non-isomorphic, an expressive algorithm produces distinct embeddings $\varphi(G_1) \neq \varphi(G_2)$ as it captures their structural differences. An edge-private algorithm, instead, protects the presence or absence

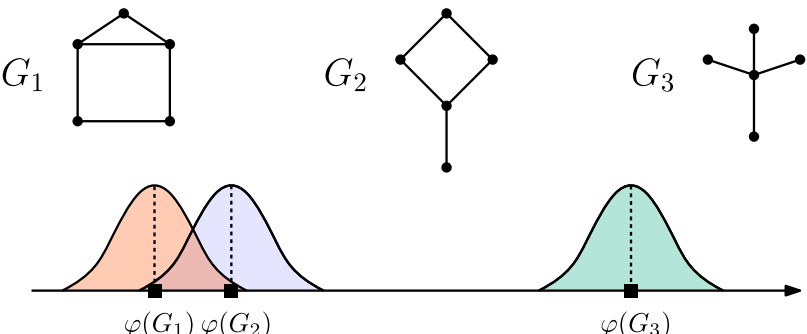

Figure 1: $G_1$ and $G_2$ are two non-isomorphic graphs that differ in exactly one edge. An *expressive* graph algorithm should distinguish between these graphs and provide different embeddings $\varphi(G_1) \neq \varphi(G_2)$; a *differentially private* algorithm instead ensures that $\varphi(G_1) \approx \varphi(G_2)$.

---

[*]Equal contribution, order is randomized. First authors are free to swap the order for professional purposes.

of individual edges and thus produces similar embeddings $\varphi(G_1) \approx \varphi(G_2)$. Therefore, requiring algorithms to be both expressive *and* private is challenging. So far, there has been little investigation towards a better theoretical understanding of this tension and a characterization of the trade-offs between privacy, expressivity, and utility, i.e., predictive performance. We fill this research gap and *investigate to which degree embeddings with provable expressivity and privacy guarantees can be obtained.* We focus on *graph-level* learning tasks while providing *edge-level* privacy guarantees. This setting allows us to study the effect of the *minimal* possible structural modifications, i.e., edge changes, on embeddings which are obtained from the graph structure only to isolate the effect of the graph structure itself on expressivity and privacy. Specifically, we consider the notions of expressivity *in expectation* and of *differential privacy*. We propose graph embeddings with carefully scaled random noise, such that their distributions sufficiently overlap for graphs that differ by one edge (see $G_1$ and $G_2$), while remaining distinguishable for graphs with larger edge edit distance (e.g., $G_1$ and $G_3$). We build upon existing work that relies on homomorphism counts, either as standalone graph representations or to increase the expressive power of graph neural networks (GNNs) (Nguyen & Maehara, 2020; Welke et al., 2023; Jin et al., 2024; Maehara & NT, 2024). Homomorphism counts are a powerful theoretical tool to investigate expressivity, as they can be used to distinguish any pair of non-isomorphic graphs (Lovász, 1967; 2012). We introduce noisy homomorphism *densities*, i.e., normalized homomorphism counts with additive noise, to obtain representations which are expressive in expectation and differentially private. Our method allows for the private release of the graph embeddings, which can be then used for any further downstream analysis.

**Main contributions.**

(i) We propose homomorphism densities as a theoretical tool to investigate the trade-off between expressivity and privacy in graph representation learning.

(ii) We show that the choice of the pattern class used to compute the homomorphism densities determines the required level of noise needed for privacy guarantees: pattern classes that provide more expressive power require more noise, which can decrease their utility.

(iii) We provide a general framework to obtain graph embeddings that satisfy a specified privacy guarantee and level of expressivity in expectation, and can be used for downstream tasks such as graph classification or regression. Our embeddings match, in expectation, the expressive power of various GNN architectures such as message-passing GNNs and subgraph GNNs, while also satisfying differential privacy.

(iv) We empirically demonstrate the expressivity-privacy trade-off and perform experiments on synthetic as well as real-world benchmark datasets. We show that our private embeddings retain good performance with increased resilience to privacy attacks.

## 2 RELATED WORK

Recent work in graph representation learning has studied the *expressive power* of learning algorithms, i.e., their ability to learn different representations for non-isomorphic graphs. A large body of work analyzed the expressive power of GNNs through the lens of $k$-Weisfeiler-Leman ($k$-WL) tests, a hierarchy of increasingly expressive color refinement algorithms (Xu et al., 2019; Morris et al., 2019). An alternative approach is to rely on graph representations built using homomorphism counts (Böker, 2021; Lovász, 2012; Jin et al., 2024; Maehara & NT, 2024; Beaujean et al., 2021; Wolf et al., 2023) to obtain arbitrarily expressive representations, at least in expectation (Nguyen & Maehara, 2020; Welke et al., 2023). Recently, Zhang et al. (2024a) and Xu (2025) have formalized a connection between homomorphism counts and the expressive power of many popular GNN architectures. While expressivity analysis can identify the theoretical limitations of learning algorithms, there is little research on how expressive power affects other properties such as, e.g., generalization, as recently pointed out by Morris et al. (2024), robustness (Campi et al., 2023; Kummer et al., 2025), or *privacy*. The lack of research on the interplay between expressivity and privacy has also been recently highlighted by Sajadmanesh et al. (2023), who call for more investigation on the expressive power of differentially private graph learning algorithms. A line of research in graph privacy focuses specifically on protecting the structural information in graphs, which is often of sensitive nature. Privacy attacks can target the edges (Raskhodnikova & Smith, 2016) or the nodes (Kasiviswanathan et al., 2013; Xiang et al., 2024) of a graph, which can encode sensitive information (Mueller et al., 2022; Li et al., 2023; Zhang et al., 2024b; Fu et al., 2023). Graph re-

construction attacks can effectively recover private information from trained models (Zhang et al., 2022; Wu et al., 2024; Zhou et al., 2023; Olatunji et al., 2023) and a number of differentially private graph learning approaches have therefore been proposed (Sajadmanesh & Gatica-Perez, 2024; 2021; Sajadmanesh et al., 2023; Pei et al., 2024; Olatunji et al., 2024; Joshi et al., 2024). To address the protection of the edges of graphs, Hidano & Murakami (2024), Xie et al. (2025), and Xu et al. (2024) focus on edge-level privacy. In particular, Hidano & Murakami (2024) consider edge-level privacy for graph-level tasks, which matches the problem setting we focus on. Furthermore, recent work has considered the problem of private subgraph counting, with a focus on triangle counting (Ding et al., 2018; Imola et al., 2022; Nguyen et al., 2023). Although expressivity and privacy have been independently studied extensively for graph learning algorithms, their interplay has not been formally investigated so far. The cited differentially private graph learning algorithms, in fact, aim to obtain the best utility under privacy constraints but do not provide expressivity guarantees. We initiate the joint study of expressivity and privacy in graph representation learning to provide a better theoretical and practical understanding of the trade-off between the two.

## 3 PRELIMINARIES

In this section, we introduce the relevant preliminaries on graph theory, expressivity, and differential privacy. More details can be found in Appendix A.

### 3.1 GRAPH THEORY AND EXPRESSIVITY IN GRAPH LEARNING

Let $\mathcal{G}$ denote the set of finite, simple graphs. A graph $G = (V, E) \in \mathcal{G}$ consists of node set $V(G)$ with $n = |V(G)|$ and edge set $E(G)$ with $e(G) = |E(G)|$. With a slight abuse of notation, let $G \setminus U$ denote the graph obtained by deleting the nodes in $U \subseteq V(G)$. For two sets $S, T \subseteq V(G)$, let $e_G(S, T)$ denote the number of edges with one endpoint in $S$ and one endpoint in $T$. For a graph $G$ with adjacency matrix $A_G$, let $\|A_G\|_1 = \sum_{i,j}^n |A_{ij}|$ denote the $\ell_1$ norm of $A_G$. A *tree decomposition* of a graph $G$ consists of a tree $T$ and a family $B = \{b_i \mid i \in V(T)\}$ of subsets of $V$ such that $(i) \bigcup_i b_i = V(G)$, $(ii)$ for every edge $uv \in E(G), \exists b_i \in B$ such that $u \in b_i$ and $v \in b_i$, and $(iii) \forall b_i, b_j, b_k$ such that $b_j$ lies on the path from $b_i$ to $b_k$, then if node $v \in b_i$ and $v \in b_k$ this implies that $v \in b_j$. The *treewidth* of a tree decomposition is $\max_i |b_i| - 1$. The treewidth of a graph $G$ is the minimum treewidth among all possible tree decompositions of $G$. Intuitively, the treewidth of a graph measures how tree-like a graph is, e.g., trees have treewidth 1 and cycles have treewidth 2. Given two graphs $F, G$, a *homomorphism* from $F$ to $G$ is an edge-preserving map $\psi : V(F) \to V(G)$. We call $\psi$ an *isomorphism* in case it is adjacency-preserving and bijective. For two graphs $G, G' \in \mathcal{G}$, let $G \simeq G'$ denote that the two graphs are *isomorphic*.

**Definition 3.1** (Homomorphism density). Let $\mathrm{hom}(F, G)$ denote the number of homomorphisms from $F$ to $G$. Then, we define the homomorphism density as

$$t(F, G) = \frac{\mathrm{hom}(F, G)}{|V(G)|^{|V(F)|}}.$$

We refer to a graph $F \in \mathcal{F} \subseteq \mathcal{G}$ as a *pattern* of the graph class $\mathcal{F}$ when we count the homomorphisms from $F$ to some graph $G$. For a given vector of patterns $\boldsymbol{F} = (F_1, \ldots, F_d)$ we consider the homomorphism density vector $\boldsymbol{t}(\boldsymbol{F}, G) := (t(F_1, G), \ldots, t(F_d, G))$. We now present two common distance notions for graphs, the edge edit distance and the cut distance.

**Definition 3.2** (Edge edit distance and cut distance, Lovász 2012; Grohe 2020). For graphs $G, G'$ with the same number of nodes, the edge distance $d_{\mathrm{edge}}$ and the cut distance $d_\square$ are defined as

$$d_{\mathrm{edge}}(G, G') = \frac{1}{2} \|A_G - A_{G'}\|_1, \qquad d_\square(G, G') = \max_{S, T \subseteq V(G)} \frac{|e_G(S, T) - e_{G'}(S, T)|}{n^2}.$$

It holds that $d_\square(G, G') \le 2d_{\mathrm{edge}}(G, G')/n^2$ (Lovász, 2012). The *counting lemma* upper bounds the absolute difference in the homomorphism densities of two graphs with respect to a pattern $F$.

**Lemma 3.1** (Counting Lemma, Lovász 2012). *For any three simple graphs $F$, $G$, and $G'$ with $|V(G)| = |V(G')|$, it holds that $|t(F, G) - t(F, G')| \le e(F)d_\square(G, G')$.*

As presented by Lovász (2012, Lemma 10.22), the counting lemma relies on a slightly different notion of the cut distance which allows to consider graphs with node sets of different cardinalities, which are not relevant for our discussion. We provide further details in Appendix A.

The expressive power of graph learning algorithms is commonly measured as their ability to distinguish between pairs of non-isomorphic graphs. Let $\varphi : \mathcal{G} \to \mathbb{R}^d$ be a *graph embedding*. We assume $\varphi$ to be permutation-invariant, i.e., for all $G, G' \in \mathcal{G}$, $G \simeq G'$ implies $\varphi(G) = \varphi(G')$. This is trivially true for homomorphism counts and homomorphism densities.

The ability of an embedding to distinguish non-isomorphic graphs is referred to as *completeness*, which we introduce as follows.

**Definition 3.3** (Completeness). An embedding $\varphi : \mathcal{G} \to \mathbb{R}^d$ is *complete* if for all $G, G' \in \mathcal{G}$, $G \simeq G'$ if and only if $\varphi(G) = \varphi(G')$.

A seminal result by Lovász asserts that homomorphism counts enjoy strong distinguishing properties, as any pair of non-isomorphic graphs can be distinguished by counting homomorphisms.

**Theorem 3.4** (Expressivity of homomorphism counts, Lovász 1967). *For any two graphs $G, G'$ it holds that $G \simeq G'$ if and only if $\hom(F, G) = \hom(F, G')$ for all simple graphs $F$.*

The embedding built from the homomorphism counts for *all* patterns $F \in \mathcal{G}$ is therefore complete. For patterns restricted to some specific graph class $\mathcal{F} \subseteq \mathcal{G}$, we introduce the following notion.

**Definition 3.5** ($\mathcal{F}$-expressivity). An embedding $\varphi : \mathcal{G} \to \mathbb{R}^d$ is $\mathcal{F}$-expressive if, for all $G, G' \in \mathcal{G}$ and for all $F \in \mathcal{F}$, $\hom(F, G) = \hom(F, G')$ if and only if $\varphi(G) = \varphi(G')$.

Consider now a random embedding, parametrized by a random variable $X \sim \mathcal{D}$ for some distribution $\mathcal{D}$ and denote it by $\varphi_X : \mathcal{G} \to \mathbb{R}^d$. We introduce notions of completeness and expressivity *in expectation* as follows.

**Definition 3.6** (Expectation-completeness, Welke et al. 2023). An embedding $\varphi_X : \mathcal{G} \to \mathbb{R}^d$ is expectation-complete if the embedding $\mathbb{E}_X[\varphi_X]$ is complete.

**Definition 3.7** ($\mathcal{F}$-expectation-expressivity). An embedding $\varphi_X : \mathcal{G} \to \mathbb{R}^d$ is $\mathcal{F}$-expectation-expressive if the embedding $\mathbb{E}_X[\varphi_X]$ is $\mathcal{F}$-expressive.

## 3.2 Differential privacy

Differential privacy (DP) is a formal notion of privacy that protects individual training points. DP is defined in terms of *neighboring databases*. A *database* is a collection of *points*, where a point in a database may be, e.g., a row in a table or an edge in a graph. Two databases $x, x'$ are *neighboring* if they differ in a single point, that is, if one single point is present in one database but not in the other. We denote this as $x \sim_1 x'$. DP guarantees that an attacker cannot confidently determine from which of two neighboring databases the output of a DP *mechanism* has been obtained from. We introduce two notions of DP and briefly describe how to achieve DP according to these notions.

**Definition 3.8** (($\epsilon, \delta$)-DP, Dwork et al. 2006). Let $\epsilon \geq 0$ and $\delta \in [0, 1)$. A randomized mechanism $\mathcal{M} : \mathcal{X} \to \mathcal{Y}$ satisfies $\delta$-approximate $\epsilon$-indistinguishability differential privacy, denoted as ($\epsilon, \delta$)-DP, if, for all neighboring $x, x' \in \mathcal{X}$ and for all $S \subseteq Range(\mathcal{M})$ it holds that $\Pr[\mathcal{M}(x) \in S] \leq e^\epsilon \Pr[\mathcal{M}(x') \in S] + \delta$, where probabilities are taken over the randomness of $\mathcal{M}$.

In DP, we refer to $\epsilon$ as the *privacy budget* of a mechanism, with larger values of $\epsilon$ providing less privacy, and a value of $\epsilon = 0$ providing perfect privacy. To make a given function $f$ DP, one can add noise proportional to its *global sensitivity* $GS_f = \max_{x \sim_1 x'} \|f(x) - f(x')\|$; see Appendix A.2 for more details. A distributional flavor of DP can be formalized in terms of the divergence of a randomized mechanism when applied to two neighboring databases.

**Definition 3.9** (($\rho, \omega$)-tCDP, Bun et al. 2018). Let $\rho > 0$ and $\omega > 1$. Let $D_\alpha(\cdot \| \cdot)$ denote the Rényi divergence of order $\alpha$ (Rényi, 1961; Van Erven & Harremos, 2014). A randomized mechanism $\mathcal{M} : \mathcal{X} \to \mathcal{Y}$ satisfies $\omega$-truncated $\rho$-concentrated differential privacy, denoted as ($\rho, \omega$)-tCDP, if, for all neighboring $x, x' \in \mathcal{X}$, for all $\alpha \in (1, \omega)$ it holds that $D_\alpha(\mathcal{M}(x) \| \mathcal{M}(x')) \leq \rho\alpha$.

Definition 3.8 and Definition 3.9 can be formally related as tCDP implies ($\epsilon, \delta$)-DP (see Lemma A.2 in Appendix A.2). It is convenient to consider tCDP as, in contrast to the standard mechanisms

described in Appendix A.2, it allows to achieve DP while considering a *local* notion of sensitivity for a function $f$ at a point $x$.

**Theorem 3.10.** *(tCDP with Gaussian noise, Bun et al. 2018) Let $f, g : \mathcal{X} \to \mathbb{R}$ satisfy, for every pair of neighboring databases $x, x' \in \mathcal{X}$ and for $\Delta_f, \Delta_g \geq 0$,*

$$|f(x) - f(x')| \leq \Delta_f \cdot e^{g(x)/2}, \quad |g(x) - g(x')| \leq \Delta_g.$$

*Let $\mathcal{M} : \mathcal{X} \to \mathbb{R}$ be the randomized mechanism defined as $\mathcal{M}(x) = f(x) + \mathcal{N}(0, e^{g(x)})$. Then, $\mathcal{M}$ satisfies $(\Delta_f^2 + \Delta_g^2, \frac{1}{2\Delta_g})$-tCDP.*

In Theorem 3.10, $\Delta_f \cdot e^{g(x)/2}$ is a *smooth* upper bound on the *local sensitivity* of $f$ at $x$. This is consistent with the smooth sensitivity framework introduced by Nissim et al. (2007), which we describe in more detail in Section 4.2 and Appendix A.2. We extend Theorem 3.10 to the $d$-dimensional case in Theorem B.6 in Appendix B.3.

## 4 EXPRESSIVITY-PRIVACY TRADE-OFF

In this section, we study the interplay between expressivity and privacy from a theoretical perspective. As previously discussed (see Figure 1), expressive embeddings can, by definition, *not* be private with respect to neighboring graphs. In the other direction, DP requires to add enough noise to mix the representations of neighboring graphs so that they may not be distinguishable, hindering expressivity. Despite this tension, we can rely on the simple observation that DP noise has mean zero to note that DP preserves embeddings *in expectation*. In this context, we take advantage of the fact that homomorphism counts can be used to obtain embeddings which are, in expectation, complete (Lovász, 2012; Nguyen & Maehara, 2020; Welke et al., 2023). Expectation-complete embeddings are a prime candidate for our analysis as they have, in expectation, arbitrary expressive power which can surpass the limits of, e.g., the WL hierarchy. We show that the *noisy* homomorphism densities, i.e., a private version of the normalized homomorphism counts, retain expressivity in expectation. To obtain embeddings which are not only private and expressive in theory but also useful in practice, we then discuss a *smooth* sensitivity bound and use this to provide formal tCDP guarantees for the embeddings. Our analysis identifies a key trade-off between expressivity and privacy: homomorphism densities obtained from patterns which are sampled from a graph class $\mathcal{F}$ that provides stronger distinguishing power require larger amounts of noise to be DP, which may practically result in worse utility for the embeddings. The graph embeddings can then be used for any downstream graph learning task without incurring further privacy cost, thanks to the post-processing property of DP (Dwork et al., 2014). We defer all missing proofs to Appendix B.

### 4.1 EXPRESSIVITY IN EXPECTATION

In this section, we show that homomorphism density vectors with DP noise are, in expectation, expressive. For now we consider a generic noise term $\mathcal{N}$ with mean zero, a condition that DP noise satisfies, and defer the precise expression for the DP noise to the next section (Section 4.2). For some graph $G$ and pattern $F$, we define the noisy homomorphism density embedding as $\tilde{t}(F, G) = t(F, G) + \mathcal{N}$. We define the noisy homomorphism density embedding $\tilde{\boldsymbol{t}}(\boldsymbol{F}, G)$ for a *vector* of patterns $\boldsymbol{F}$ analogously. Note that $\tilde{\boldsymbol{t}}(\boldsymbol{F}, G)$ is *not* permutation-invariant due to the added noise, a necessary consequence of the fact that DP requires a *randomized* mechanism. This observation does not, however, affect the possibility to obtain expressive or even complete graph embeddings *in expectation*.[1] For our results, similarly to Welke et al. (2023), we assume that each pattern is sampled from an appropriate distribution $\mathcal{D}$ with full support on the graph class $\mathcal{F}$ of interest.

We first show that for any fixed graph and a single sampled pattern, the noisy homomorphism density embedding is expressive in expectation.

**Theorem 4.1.** *For any $G \in \mathcal{G}$, $\tilde{t}(F, G)$ is $\mathcal{F}$-expectation-expressive for $F \sim \mathcal{D}$ if $\mathcal{D}$ has full support on $\mathcal{F} \subseteq \mathcal{G}$. If $\mathcal{F} = \mathcal{G}$, then $\tilde{t}(F, G)$ is expectation-complete.*

---

[1] Note that homomorphism densities, in contrast to homomorphism counts, do not distinguish $G$ and a *blowup* of $G$. We discuss this issue and a simple solution to it at the end of this section as well as in Appendix B.

As we are often interested in a homomorphism density *vector* obtained from a number of sampled patterns, we extend Theorem 4.1 to the vector case. We show that the resulting noisy homomorphism density embedding is not only expressive in expectation, but also remains expressive with high probability for a large enough number of sampled patterns.

**Theorem 4.2.** *Let $\mathcal{D}$ be a distribution on $\mathcal{F} \subseteq \mathcal{G}$ with full support. Let $G \in \mathcal{G}$, $\mathbf{F} \sim \mathcal{D}^d$, and $\theta \in [0, 1]$. For large enough $d$, $\tilde{\mathbf{t}}(\mathbf{F}, G)$ is $\mathcal{F}$-expressive with probability at least $1 - \theta$. If $\mathcal{F} = \mathcal{G}$, then, for large enough $d$, $\tilde{\mathbf{t}}(\mathbf{F}, G)$ is complete with probability at least $1 - \theta$.*

Theorem 4.1 and Theorem 4.2 demonstrate that, despite the noise required for DP, our homomorphism density embeddings retain full discriminative power in expectation and, with enough patterns, with high probability.

## 4.2 PRIVACY GUARANTEES

In this section, we provide DP guarantees for the homomorphism density embeddings. To calibrate an appropriate amount of noise to be added to the homomorphism densities to guarantee DP, we discuss how to bound the sensitivity of $\mathbf{t}(\mathbf{F}, G)$. In particular, we choose tCDP as our formal notion of privacy since it allows us to consider a *local* notion of sensitivity with Gaussian noise, which often requires less noise to be added in practical settings. In most of the following discussion, we consider the pattern $F$ to be fixed, but we remark that to achieve expressivity in expectation (see Section 4.1) the patterns are sampled from a distribution as $F \sim \mathcal{D}$. We focus on *edge-level* privacy and strive to protect the presence/absence of individual edges in a graph. We thus interpret neighboring graphs, according to the following definition, as two neighboring databases.

**Definition 4.3** (Neighboring graphs). *Two graphs $G$, $G'$ with the same number of nodes are neighboring graphs, written $G \sim_1 G'$, if $d_{\text{edge}}(G, G') = 1$.*

Based on our notion of neighboring graphs, we can leverage the counting lemma to obtain a bound on the *global sensitivity* of the homomorphism densities: For any two neighboring graphs $G \sim_1 G'$ with $n$ nodes and for any pattern $F$ we get that $GS_{t,F} = |t(F, G) - t(F, G')| \le e(F)d_\square(G, G') = 2e(F)/n^2$ (see Corollary B.1). Since $GS_{t,F}$ considers the worst case behavior of $t$ around any graph $G$, using it in the standard DP mechanism (Appendix A.2) is likely to result in poor performance[2]. In contrast, local sensitivity, defined as $LS_{t,F}(G) = \max_{G' \in \mathcal{G}: d_{\text{edge}}(G, G') \le 1} |t(F, G) - t(F, G')|$ provides an upper bound on the sensitivity *around a specific graph* $G$, and is often much smaller than $GS_{t,F}$. Additive noise proportional to the local sensitivity, however, does not guarantee DP. A crucial step in our analysis is therefore to consider noise calibration under the smooth sensitivity framework (Nissim et al., 2007), which provides a *smooth* upper bound to the local sensitivity. For some $\beta > 0$ and pattern $F$, the $\beta$-*smooth* sensitivity of $t(F, G)$ at $G$ is defined as

$$S_{t,F}(G) = \max_{G' \in \mathcal{G}} \left( e^{-\beta d_{\text{edge}}(G, G')} \cdot LS_{t,F}(G') \right). \tag{1}$$

As we consider homomorphism density *vectors*, we show in our next proposition how to provide an upper bound to the smooth sensitivity of $\mathbf{t}(\mathbf{F}, G)$ by considering the individual $S_{t,F_i}(G)$ for $F_i \in \mathbf{F}$.

**Proposition 4.4.** *Let $S_{t,*}(G) = \|S_{t,F_1}(G), \dots, S_{t,F_d}(G)\|_2$ and $\beta > 0$. Let*

$$S_t(G) = \max_{H \in \mathcal{G}} \left( e^{-\beta d_{edge}(G, H)} \max_{H' \in \mathcal{G}: d_{edge}(H, H') \le 1} \|\mathbf{t}(\mathbf{F}, H) - \mathbf{t}(\mathbf{F}, H')\|_2 \right) \tag{2}$$

*be the $\beta$-smooth sensitivity of $\mathbf{t}(\mathbf{F}, G)$ at $G$. Then, it holds that $S_{t,*}(G) \ge S_t(G)$.*

In many cases, domain knowledge allows to assume that the degree of the graphs is bounded. We thus derive an even smaller bound on the sensitivity of the homomorphism densities.

**Theorem 4.5** (Sensitivity of homomorphism density for bounded degree graphs). *Let $G \sim_1 G'$ be two neighboring graphs with $n$ nodes and maximum degree $\Delta_{\max}$. For any pattern $F$ with $m > 1$ nodes, it holds that*

$$|t(F, G) - t(F, G')| \le \frac{2e(F)}{n^2} \left( \frac{\Delta_{\max}}{n} \right)^{m-2}. \tag{3}$$

---

[2]See the ablation study in Appendix D.3 for empirical evidence supporting this claim.

For large graphs and large patterns $\left(\frac{\Delta_{\max}}{n}\right)^{m-2} \ll 1$. Therefore, the bound provided by Equation (3) is often tighter in practice than the one we could directly obtain from the counting lemma. For domains where no meaningful public degree bound is available, one could either estimate $\Delta_{\max}$ privately or, simply, set $\Delta_{\max} = n$ to recover the counting lemma. For a private estimate, one may add, e.g., Laplacian noise to the empirical maximum degree under a small additional privacy budget. For a given density vector $\boldsymbol{t}(\boldsymbol{F}, G)$, we can now use Theorem 4.5 to upper bound the smooth sensitivity of each $t(F, G)$ individually and obtain an upper bound $S_{t,*}(G)$ for the entire vector as shown in Proposition 4.4. With this, we present the main result of this section: a private mechanism for homomorphism density vectors. More specifically, we derive a tCDP version of $\boldsymbol{t}(\boldsymbol{F}, G)$.

**Theorem 4.6.** *Let $\boldsymbol{t}(\boldsymbol{F}, G)$ be the homomorphism density vector for graph $G$ and pattern set $\boldsymbol{F}$ with $|\boldsymbol{F}| = d$, $\rho' > 0$, and let $S_{t,*}(G)$ be a $\beta$-smooth upper bound to the local sensitivity as per Proposition 4.4. Then, the mechanism*

$$\tilde{\boldsymbol{t}}(\boldsymbol{F}, G) = \boldsymbol{t}(\boldsymbol{F}, G) + \mathcal{N}\left(\boldsymbol{0}, \frac{[S_{t,*}(G)]^2}{2\rho'}I_d\right) \tag{4}$$

*is $\left(2\rho' + d \cdot 4\beta^2, \frac{1}{4\beta}\right)$-tCDP for neighboring graphs as per Definition 4.3.*

Theorem 4.6 enables us to determine the amount of noise needed to guarantee tCDP for the homomorphism densities. The additive noise has mean zero, and thus our results on expectation-expressivity of the previous section apply: in expectation, the noisy, private homomorphism densities are unbiased with respect to the non-private densities. As smooth sensitivities are upper-bounded by the global sensitivity, we expect the mechanism described in Theorem 4.6 to yield a significantly better privacy-utility trade-off compared to the standard Gaussian mechanism; see Table 6 in Appendix D.3 for supporting empirical evidence.

### 4.3 PRIVATE AND EXPRESSIVE GRAPH REPRESENTATIONS

We are now able to combine the results of the previous two sections to $(i)$ show how to obtain provably expressive and private graph representations (Theorem 4.8), and $(ii)$ formally quantify the expressivity-privacy trade-off (Proposition 4.9). First, we highlight a technicality on how to distinguish *blowup* graphs, which we also discuss more thoroughly in Remark B.1 in Appendix B.

*Remark* 4.7 (Completeness of homomorphism density embeddings). A $p$-blowup of $G$ can be obtained by replacing each node of $G$ by $p \geq 1$ twin copies (Lovász, 2012). Two graphs $G, G'$, where $G'$ is a *blowup* of $G$, have the same homomorphism density for any pattern $F$ (Lovász, 2012, Theorem 5.32). Therefore, homomorphism densities cannot be used to distinguish all non-isomorphic graphs. To resolve this, we append the node count $|V(G)|$ to the homomorphism density embedding of $G$ to distinguish it from all its blowups. This operation is trivially DP with respect to the neighboring graph notion in Definition 4.3, as any two neighboring graphs have the same number of nodes, and thus costs no further privacy budget.

Our first result in this section states that we can generate graph embeddings which are provably private and expressive. We show that for a chosen privacy budget and a chosen graph class $\mathcal{F}$, we guarantee that our homomorphism density embeddings are tCDP and $\mathcal{F}$-expectation-expressive.

**Theorem 4.8.** *Let $\mathcal{D}$ be a distribution on $\mathcal{F} \subseteq \mathcal{G}$ with full support. Let $G \in \mathcal{G}$ be a graph and $\boldsymbol{F} = (F_1, \ldots, F_d) \sim \mathcal{D}^d$ be a vector of patterns. Then, the graph representation $\tilde{\boldsymbol{t}}(\boldsymbol{F}, G) = \boldsymbol{t}(\boldsymbol{F}, G) + \mathcal{N}\left(\boldsymbol{0}, \frac{[S_{t,*}(G)]^2}{2\rho'}I_d\right)$ is $\mathcal{F}$-expectation-expressive and $(2\rho' + d \cdot 4\beta^2, \frac{1}{4\beta})$-tCDP, where $\rho' > 0$ and $S_{t,*}(G)$ is a $\beta$-smooth upper-bound on the local sensitivity of $\boldsymbol{t}(\boldsymbol{F}, G)$. If $\mathcal{F} = \mathcal{G}$, then $\tilde{\boldsymbol{t}}(\boldsymbol{F}, G)$ is also expectation-complete.*

Theorem 4.8 allows us to characterize the expressive power of our embeddings more precisely by sampling patterns from a graph class $\mathcal{F}$ that determines a certain level of expressivity in expectation (Nguyen & Maehara, 2020). For instance, it is well known that 1-WL serves as upper bound for the expressive power of a large class of message-passing graph neural networks (MPNNs) (Xu et al., 2019; Morris et al., 2019). The expressive power of 1-WL, in turn, is equivalent to counting *tree* homomorphisms. In other words, two graphs have the same 1-WL color multiset (Xu et al., 2019) if and only if they have the same homomorphism counts for all trees. This equivalence can be generalized for many popular GNN architectures by determining their *homomorphism-distinguishing closed*

graph class (Neuen, 2024), which corresponds to the pattern graph class in our setting. For instance, the expressive power of $k$-GNNs (Morris et al., 2019; 2023) corresponds to the homomorphism-distinguishing closed graph classes of treewidth $k$ (Neuen, 2024). We refer to Zhang et al. (2024a) and Xu (2025) for a more in-depth discussion of homomorphism expressivity and general techniques to obtain homomorphism-distinguishing closed graph classes for given GNN architectures.

Table 1: Common GNNs and their homomorphism-distinguishing closed graph classes. See Paolino et al. (2024) for $r$-$\ell$MPNNs, Gai et al. (2025) for spectral invariant GNNs, and Zhang et al. (2022) for the remaining GNNs. For details on the maximum numbers of edges refer to Appendix B.2.

| GNN | Graph class $\mathcal{F}$ | $\max_{F \in \mathcal{F}, m=|V(F)|} e(F)$ |
|---|---|---|
| MPNNs (1-WL) | Trees | $m - 1$ |
| $r$-$\ell$MPNNs ($r$-$\ell$WL) | Fan-cactus graphs | $2m - 3$ |
| Spectral invariant GNNs | Parallel trees | $2m - 3$ |
| Subgraph $k$-GNNs | $\{F : \exists U \subset V(F) \text{ s.t. } |U| \leq k \text{ and } F \setminus U \text{ is a forest}\}$ | $m(k+1) - 1 - \frac{k^2 + 3k}{2}$ |
| $k$-FGNNs ($k$-WL) | $\{F : \mathrm{tw}(F) \leq k\}$ | $km - \frac{k(k+1)}{2}$ |

Based on our theoretical investigation, we now present our second result and quantify the trade-off between expressivity and privacy: the choice of the graph class $\mathcal{F}$ does not only affect expressivity, but also the amount of noise that needs to be added to the embeddings to obtain privacy guarantees.

**Proposition 4.9.** *Consider a graph $G$ with $n$ nodes and let $\mathcal{F}$ be a class of patterns. For a chosen privacy parameter $\rho' > 0$, the Gaussian noise necessary to obtain a specific privacy guarantee in Theorem 4.8 has variance $\sigma^2 = \mathcal{O}\left((\max_{F \in \mathcal{F}} e(F))^2 / n^4\right)$.*

Table 1 provides the homomorphism-distinguishing closed graph class $\mathcal{F}$ as well as its maximum number of edges for some well-known GNN architectures with expressive power precisely characterized by $\mathcal{F}$, i.e., that can distinguish all non-isomorphic graphs in $\mathcal{F}$. With this information, we are able to generate private graph embeddings that, in expectation, match the expressive power of many GNN architectures. In general, more expressive GNN architectures often have greater bounds on $e(F)$ for $F \in \mathcal{F}$. From Proposition 4.9 we can therefore conclude that with patterns sampled from more expressive graph classes, more noise is required to achieve a given privacy guarantee. *Thus, we have identified an explicit trade-off between expressivity and privacy.*

## 5 EXPERIMENTS

We complement our theoretical investigation with a compact empirical study that can be run on a single commercial GPU. Our goal is not to compete with state-of-the-art approaches, but rather to probe the trade-off between the desiderata of expressivity and privacy in practice, as well as assess the utility, i.e., the performance, of our private embeddings on real-world and synthetic datasets. We organize our experimental evaluation around the following research questions:

**Q1.** For a fixed privacy budget, do embeddings obtained from more expressive graph classes offer better performance?

**Q2.** For a fixed class of patterns, how does performance degrade when we require stronger privacy guarantees?

**Q3.** Are the expressive and private embeddings we obtain practically useful?

**Experimental setup.** We evaluate the private and expressive homomorphism density vectors for graph-level tasks on real-world as well as synthetic datasets. We run experiments on four commonly used OGBG molecular benchmark datasets (MOLHIV, MOLBACE, MOLBBBP, and MOLLIPO (Hu et al., 2020)) and on three network datasets (REDDIT-BINARY and REDDIT-MULTI-5K (Xiang et al., 2024), and GitHub STARGAZERS (Rozemberczki et al., 2020)). Additionally, we perform experiments on a synthetic stochastic block model (SBM, see Appendix C.2). As we focus on how the graph structure can be privately leveraged to have expressive representations, the core of our experiments relies on the graph structure *only* as encoded by the homomorphism density vectors

and does not consider any node or edge features. We experiment with different privacy parameters $\rho' \in [10^{-8}, 1]$. To make results more interpretable, we convert our tCDP guarantees into $(\epsilon, \delta)$-DP guarantees using Lemma A.2 in Appendix A.2; $(\epsilon, \delta)$-DP guarantees are easier to interpret as privacy budgets roughly in the range $\epsilon \in (0, 10]$ are generally understood to provide meaningful privacy protection in graph machine learning (Wu et al., 2022; Sajadmanesh & Gatica-Perez, 2021). To evaluate privacy protection empirically, we run the privacy attacks detailed in Appendix D to try and recover the original graphs. We consider sampled vectors of patterns $F$ with $d = 50$ for all experiments. We compare the performance of our embeddings against the Randomized Response (RR) and the degree-preserving Randomized Response (DPRR, Hidano & Murakami, 2024) GNN baselines. For details on the experimental setup, baselines, and additional results, see Appendix D.

**Time complexity.** While counting homomorphisms is intractable in general, there exist efficient algorithms for certain graph classes. For instance, homomorphism counts for cycle patterns can be computed efficiently via powers of the adjacency matrix, see Proposition C.2 in Appendix C. For bounded treewidth patterns, Díaz et al. (2002) introduced a polynomial-time algorithm that computes $\mathrm{hom}(F, G)$ in $\mathcal{O}(|V(F)| \, |V(G)|^{\mathrm{tw}(F)+1})$, where $\mathrm{tw}(F)$ denotes the treewidth of pattern $F$. Based on this result, Welke et al. (2023) devise a sampling strategy with polynomial runtime in expectation by decreasing the probability mass of patterns with higher treewidth. The key idea behind their approach is to construct a probability distribution and ensure that every pattern has a nonzero, but potentially very small probability to be sampled. We refer to Welke et al. (2023, Theorem 15, Appendix C) for a more in-depth discussion on the construction of such a distribution and details on the sampling strategy. We remark that the computation and addition of DP noise does not introduce any noticeable computational overhead. Furthermore, the computation of our embeddings can be regarded as a one-time pre-processing step and they can subsequently be used for any downstream analysis. In practice, counting homomorphisms from bounded treewidth patterns, which matches the expressive power of highly expressive GNNs, can be done on standard consumer-grade hardware; see Table 7 in Appendix D.4 for detailed runtimes for MOLHIV with increasing maximum treewidth of $\mathrm{tw} = \{1, 2, 3\}$ and for REDDIT-BINARY with $\mathrm{tw} = 1$.

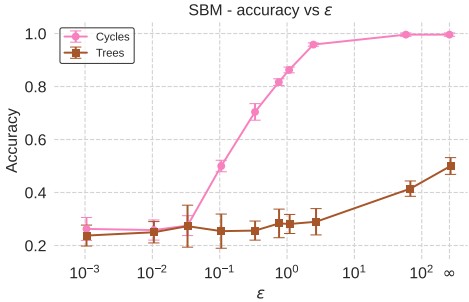
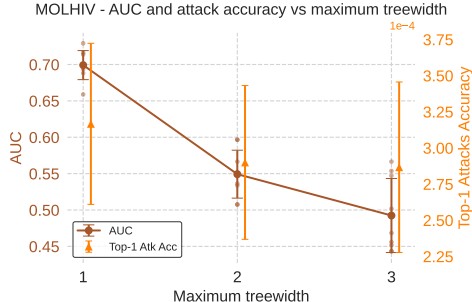

(a) Classification accuracy vs privacy budget $\epsilon$ on SBM with $k$-NN for tree and cycle patterns.

(b) Classification AUC and attack accuracy vs max treewidth on MOLHIV with $k$-NN for $\epsilon = 1$.

Figure 2: Visualizations for two of our experiments on SBM and MOLHIV. We report average results with error bars of 2 standard deviations across 9 runs.

**Results.** To answer questions **Q1** and **Q2** we perform two sets of experiments: In the first set of experiments (a), we consider a learning task where one pattern class is provably more expressive than another pattern class. In the second set of experiments (b), we instead consider a learning task where more expressive patterns are not expected to improve performance. For (a), we construct an SBM dataset, for which cycle patterns have provably stronger distinguishing properties than tree patterns (see Lemma C.1 in Appendix C.2). We observe in Figure 2a that cycle homomorphism densities, a pattern class that ensures expressivity in expectation, result in drastically better practical performance compared to tree homomorphism densities. We can further see that the performance for cycles remains good for reasonable privacy budgets around $\epsilon = 1$. For (b), we use patterns with increasing maximum treewidth of $\{1, 2, 3\}$, and therefore increasing expressive power, on MOLHIV. On this dataset, patterns with maximum treewidth of $1$ achieve good performance. In this case, choosing patterns from more expressive graph classes may not offer a benefit in performance: to

Table 2: Utility and attack accuracy for our experiments on OGBG datasets. As utility metric, we use the regression RMSE for MOLLIPO and the classification AUC for MOLHIV, MOLBBBP, and MOLBACE. We report average results and standard deviations across 9 runs. **Bold** marks best results for the private runs.

| $t(\boldsymbol{F}, G)$ | | MOLHIV ↑ | MOLBBBP ↑ | MOLBACE ↑ | MOLLIPO ↓ |
|---|---|---|---|---|---|
| Private ($\epsilon = 1$) | Utility | **0.692** (0.020) | **0.602** (0.005) | **0.652** (0.069) | **1.086** (0.004) |
| | Attack | 0.003 (< 0.001) | 0.025 (0.003) | 0.027 (0.002) | 0.011 (0.002) |
| Non private ($\epsilon = \infty$) | Utility | 0.745 (< 0.001) | 0.644 (0.008) | 0.752 (0.002) | 1.055 (0.002) |
| | Attack | 0.955 (0.020) | 1.000 (< 0.001) | 0.990 (< 0.001) | 0.992 (0.006) |
| GNN Baseline | | | | | |
| Private ($\epsilon = 1$) RR | Utility | 0.488 (0.008) | 0.440 (0.005) | 0.457 (0.024) | 1.568 (0.248) |
| Private ($\epsilon = 1$) DPRR | Utility | 0.595 (0.155) | 0.539 (0.019) | 0.648 (0.043) | 1.499 (0.333) |
| Non private ($\epsilon = \infty$) | Utility | 0.672 (0.022) | 0.586 (0.027) | 0.768 (0.033) | 1.033 (0.021) |

obtain the same privacy guarantee and comparable resilience to privacy attacks, we expect to add more noise to more expressive patterns according to Proposition 4.9. In fact, in Figure 2b, we observe a downward trend in the AUC as we increase the maximum treewidth. This confirms that *there is indeed a practical trade-off between expressivity and privacy.* Finally, our results in Table 2 and Table 4 (in Appendix D.2) positively answer **Q3**: overall, our private and expressive homomorphism density vectors are useful embeddings for graph classification and regression tasks. Moreover, the private embeddings offer significantly better resilience to privacy attacks compared to their non-private counterparts. For a reasonable privacy budget of $\epsilon = 1$, we always outperform the baselines on the OGBG datasets and MOLHIV, MOLBBBP, and MOLLIPO stay within 90% of the performance of the non-private homomorphism density embeddings. For the network datasets, we obtain performances comparable to those in Hidano & Murakami (2024), despite relying on significantly simpler classifiers, which further confirms the practical usefulness of our embeddings; see Appendix D for a more in-depth discussion. Indeed, we emphasize that for all our experiments we use basic machine learning algorithms such as $k$-nearest neighbor ($k$-NN) classifiers, support vector machines, and random forests (see Appendix D.1). This suggests that *the private embeddings we obtain are themselves highly informative.* We report further experiments and ablation studies in Appendix D, where we verify the impact of node features on utility, and we confirm that a straightforward implementation of DP with global sensitivity yields unusably noisy embeddings. This further highlights the necessity for the more refined bounds on sensitivity we discuss in Section 4.2 to obtain practically useful private embeddings.

## 6 CONCLUSION

We study the trade-off between expressivity and privacy in graph representation learning. Our results first address an existing research gap on the interplay between the desiderata of expressivity and privacy. We propose a noisy version of homomorphism densities as graph embeddings, and show that our embeddings satisfy formal expressivity and differential privacy guarantees. Our expressive and private embeddings can be used for any downstream task and with any machine learning algorithm without incurring further privacy costs. In our experiments, we show that our embeddings are useful in practice and retain high performance with practical protection against privacy attacks.

**Limitations.** A natural limitation of our approach is that it inherits all the limitations of homomorphism counts and densities. As discussed, homomorphism counts *can* be expensive to compute. Moreover, homomorphism counts alone may not be sufficient for good practical performance, especially if node features and their topological arrangements are crucial for the task at hand.

**Future work.** A promising direction for future work is to refine the noise calibration by more precisely analyzing the sensitivity of specific graph classes, and to privately encode edge features to further improve the privacy–utility trade-off.

ETHICS STATEMENT

Our work provides formal guarantees that align with the increasing regulatory push toward privacy-preserving machine learning models. We next detail our usage of LLMs. We use LLMs for the following use cases: $(i)$ as a coding assistant, $(ii)$ for discussion and suggestions on the experimental setup, $(iii)$ for retrieval and discovery of related work, and $(iv)$ for feedback on the final draft of our submission. In all cases, we, the authors, were the last ones to check and modify the content accordingly. Furthermore, we ensured that the content of this submission was not used for further training of LLMs to not bias the reviewing process.

REPRODUCIBILITY STATEMENT

All our theoretical statements are supported by proofs, which can be found in Appendix B, and/or pointers to existing literature. Our experimental setup is detailed in Appendix D and the code used to produce our empirical results can be found here: https://github.com/tamaramagdr/private-homcounts.

AUTHOR CONTRIBUTIONS

The idea was conceived and developed jointly by PI and TD, who wrote and proofread the theoretical contribution, the code, and the manuscript under the supervision of TG.

ACKNOWLEDGMENTS

We would like to thank Maximilian Thiessen, Pascal Welke, and Fabian Jogl for helpful discussions and feedback. We would also like to thank Flaviano De Santis and Christoph Sandrock for proof-reading our submission. This work was funded in part by the Vienna Science and Technology Fund (WWTF), project StruDL (ICT22-059); by the Austrian Science Fund (FWF), project NanOX-ML (6728); by the TU Wien DK SecInt and by the European Unions Horizon Europe Doctoral Network programme under the Marie-Skłodowska-Curie grant, project Training Alliance for Computational systems chemistry (101072930).

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

# A   ADDITIONAL PRELIMINARIES

In this section we provide additional details on the preliminaries.

## A.1   CUT NORM

In our preliminaries we have implicitly assumed that $G$ and $G'$ are defined on the same node set, i.e., the nodes of $G$ and $G'$ have some fixed labeling which minimizes the cut distance. If two graphs $G$ and $G'$ have the same cardinality $n$ but different node sets, their distance is defined as

$$\hat{\delta}_\square(G, G') = \min_{\hat{G}, \hat{G}'} d_\square(\hat{G}, \hat{G}'), \tag{5}$$

with $\hat{G}$ and $\hat{G}'$ ranging over all possible labelings of the nodes of $G$ and $G'$.

For two graphs $G$ and $G'$ with different cardinalities, we define the cut distance using *fractional overlays*. A fractional overlay of two graphs $G$ of order $n$ and $G'$ of order $n'$ is a nonnegative $n \times n'$ matrix $X = [X_{iu}]_{n \times n'}$ such that $\sum_{u=1}^{n'} X_{iu} = \frac{1}{n}$ and $\sum_{i=1}^{n} X_{iu} = \frac{1}{n'}$. If $n = n'$, let $\sigma : V(G) \to V(G')$ be a bijection. Then, $X_{iu} = \frac{1}{n}\mathbf{1}_{\sigma(i)=u}$ is a fractional overlay. For a fixed fractional overlay $X$, we define the *labeled cut distance* as

$$d_\square(G, G', X) = \max_{Q, R \subseteq V(G) \times V(G')} \Big| \sum_{\substack{iu \in Q \\ jv \in R}} X_{iu} X_{jv} \big( \mathbf{1}_{ij \in E(G)} - \mathbf{1}_{uv \in E(G')} \big) \Big|.$$

The cut distance between $G$ and $G'$ is defined over all overlays $\mathcal{X}(G, G')$:

$$\delta_\square(G, G') = \min_{X \in \mathcal{X}(G,G')} d_\square(G, G', X). \tag{6}$$

Note that, in general, for two graphs with the same cardinality $\delta_\square$ may not coincide with $\hat{\delta}_\square$ and it holds that $\delta_\square(G, G') \leq \hat{\delta}_\square(G, G')$ (Lovász, 2012). We can now re-state the counting lemma with more precise notation.

**Lemma A.1** (Counting Lemma Lovász 2012, Lemma 10.22)**.** *For any three simple graphs $F$, $G$, and $G'$, it holds that:*

$$|t(F, G) - t(F, G')| \leq e(F)\delta_\square(G, G'). \tag{7}$$

As in our setting we consider pairs of graphs $G, G'$ with the same number of nodes which share the same node set, we have that $d_\square(G, G') = \delta_\square(G, G')$ and we thus do not need to consider the cut distance defined over fractional overlays.

## A.2   DIFFERENTIAL PRIVACY

We provide here additional preliminaries on DP, with a focus on how to achieve DP with additive noise scaled to the *global* sensitivity of a function.

**Definition A.1** ($\epsilon$-DP, Dwork 2006)**.** Let $\epsilon \geq 0$. A randomized mechanism $\mathcal{M} : \mathcal{X} \to \mathcal{Y}$ satisfies $\epsilon$-indistinguishability differential privacy, denoted as $\epsilon$-DP, if, for all neighboring $x, x' \in \mathcal{X}$ and all $S \subseteq Range(\mathcal{M})$ it holds that

$$\Pr[\mathcal{M}(x) \in S] \leq e^\epsilon \Pr[\mathcal{M}(x') \in S], \tag{8}$$

where probabilities are taken over the randomness of $\mathcal{M}$.

**Definition A.2** (($\epsilon, \delta$)-DP, Dwork et al. 2006)**.** Let $\epsilon \geq 0$ and $\delta \in [0, 1)$. A randomized mechanism $\mathcal{M} : \mathcal{X} \to \mathcal{Y}$ satisfies $\delta$-approximate $\epsilon$-indistinguishability differential privacy, denoted as ($\epsilon, \delta$)-DP, if, for all neighboring $x, x' \in \mathcal{X}$ and all $S \subseteq Range(\mathcal{M})$ it holds that

$$\Pr[\mathcal{M}(x) \in S] \leq e^\epsilon \Pr[\mathcal{M}(x') \in S] + \delta, \tag{9}$$

where probabilities are taken over the randomness of $\mathcal{M}$.

In the literature, $\epsilon$-DP is also referred to as *pure* DP while $(\epsilon, \delta)$-DP is also referred to as *approximate* DP. Given a deterministic function $f$, one can build a private mechanism from $f$ by means of additive noise calibrated to its global sensitivity $GS_{f,p} = \max_{x \sim x'} \|f(x) - f(x')\|_p$, where $\|\cdot\|_p$ is a $\ell_p$-norm. When $p$ is omitted, we consider $\ell_2$ norms.

**Theorem A.3** (Laplace mechanism for pure DP, Dwork 2006; Dwork et al. 2014). *Let $f : \mathcal{X} \to \mathbb{R}$ have $\ell_1$ sensitivity $GS_{f,\ell_1}$. The randomized mechanism $\mathcal{M}(x) = f(x) + Lap\left(\frac{GS_{f,\ell_1}}{\epsilon}\right)$ satisfies $\epsilon$-DP, where $Lap(b)$ denotes Laplacian noise with mean $0$ and scale $b$.*

**Theorem A.4** (Gaussian mechanism for approximate DP, Dwork et al. 2006; Dwork 2006). *Let $f : \mathcal{X} \to \mathbb{R}$ have $\ell_2$ sensitivity $GS_{f,\ell_2}$. The randomized mechanism $\mathcal{M}(x) = f(x) + \mathcal{N}(0, \sigma^2)$ satisfies $(\epsilon, \delta)$-DP for $\sigma \geq \frac{GS_{f,\ell_2}\sqrt{2\ln(1.25/\delta)}}{\epsilon}$.*

**Lemma A.2** (tCDP implies $(\epsilon, \delta)$-DP, Bun et al. 2018). *Suppose mechanism $\mathcal{M}$ satisfies $(\rho, \omega)$-tCDP with a Rényi divergence of order $\alpha$. Then, for all $\delta \in [0, 1)$, $1 < \alpha \leq \omega$, $\mathcal{M}$ satisfies $(\epsilon, \delta)$-DP with*

$$\epsilon = \begin{cases} \rho + 2\sqrt{\rho \ln(1/\delta)} & if \quad \ln(1/\delta) \leq (\omega - 1)^2 \rho \\ \rho\omega + ln(1/\delta)/(\omega - 1) & if \quad \ln(1/\delta) \geq (\omega - 1)^2 \rho. \end{cases}$$

**Definition A.5** (Smooth Sensitivity, Nissim et al. 2007). For a function $f : \mathcal{X} \to \mathbb{R}$, let $d(x, x')$ measure the distance between $x$ and $x'$, where $d(x, x') = 1$ indicates that $x \sim x'$. Define the *local sensitivity* of $f$ at $x$ as

$$LS_f(x) = \max_{x' \in \mathcal{X}: d(x,x') \leq 1} |f(x) - f(x')|. \tag{10}$$

For $\beta > 0$, the $\beta$-*smooth sensitivity* of $f$ at $x$ is then defined as

$$S_f(x) = \max_{y \in \mathcal{X}} e^{-\beta \, d(x,y)} LS_f(y). \tag{11}$$

It is immediate to see that for all $x \in \mathcal{X}$, it holds that $LS_f(x) \leq GS_f$. Therefore, we expect a method that relies on smooth sensitivities to provide better utility, compared to one that relies on global sensitivities.

# B    MISSING PROOFS

In this section, we restate our theoretical results and provide proofs that we omitted in the main part.

## B.1    EXPRESSIVITY

*Remark* B.1 (On graph blowups). For the following proofs, it is necessary to address the fact that two graphs $G, G'$, where $G'$ is a *blowup* of $G$, have the same homomorphism density for any pattern $F$ (Lovász, 2012, Theorem 5.32). A $p$-blowup of $G$ can be obtained by replacing each node of $G$ by $p \geq 1$ twin copies (Lovász, 2012). Therefore, homomorphism densities cannot be used to distinguish all non-isomorphic graphs. This subtlety is not addressed in Welke et al. (2023) who, in fact, rely on the wrong assumption that homomorphism densities are complete for some of their results. We can address this in two ways. We can $(i)$ rely on homomorphism *counts*, which do not present the same problem and can be used to obtain a complete embedding (Lovász, 1967; Welke et al., 2023). As our DP statements consider pairs of graphs with the same number of nodes, this only requires to rescale the definitions of sensitivity and leads to equivalent statements about the privacy of the embeddings. This does not affect the utility of our embeddings which are, simply, rescaled. Alternatively, we can $(ii)$ append the node count $|V(G)|$ to the homomorphism density embedding of $G$ to distinguish it from all its blowups. This operation is trivially DP with respect to the neighboring graph notion in Definition 4.3 and costs no further privacy budget. As we rely on the counting lemma to derive our sensitivity bounds, we choose to present our results in terms of homomorphism densities. Therefore, we will assume that, if necessary, the node count is appended to the embedding so that the following statements hold[3]. We stress that this is simply a choice of presentation, as all our expressivity and privacy statements could be easily rephrased in terms of homomorphism counts.

---

[3]Note that this can also be used to recover the results presented in Welke et al. (2023) that rely on the completeness of homomorphism densities.

**Theorem 4.1.** *For any $G \in \mathcal{G}$, $\tilde{t}(F, G)$ is $\mathcal{F}$-expectation-expressive for $F \sim \mathcal{D}$ if $\mathcal{D}$ has full support on $\mathcal{F} \subseteq \mathcal{G}$. If $\mathcal{F} = \mathcal{G}$, then $\tilde{t}(F, G)$ is expectation-complete.*

*Proof.* Consider

$$\tau = \mathbb{E}_F[t(F, G)] = \sum_{F' \in \mathcal{F}} \Pr_{\mathcal{D}}(F = F') t(F', G) e_{F'}, \tag{12}$$

where $e_{F'} \in \mathbb{R}^{|\mathcal{F}|}$ is a standard basis unit vector of $\mathbb{R}^{|\mathcal{F}|}$. We can write $\tilde{t}(F, G) = t(F, G) + Y$ where $Y \sim \mathcal{N}(\mu_Y = 0, \sigma^2)$ for some variance $\sigma^2$. Note that $Y$ and $F$ are independent random variables. It then holds that

$$\mathbb{E}[\tilde{t}(F, G)] = \mathbb{E}[t(F, G) + Y e_F] = \mathbb{E}_F[t(F, G)] + \mathbb{E}_Y[Y e_F] \tag{13}$$
$$= \mathbb{E}_F[t(F, G)] + \mathbb{E}_Y[Y] \mathbb{E}_Y[e_F] = \mathbb{E}_F[t(F, G)] + \mu_Y \mathbb{E}_Y[e_F] \tag{14}$$
$$= \mathbb{E}_F[t(F, G)] = \tau. \tag{15}$$

It remains to show that $\tau$ is $\mathcal{F}$-expressive. Let $G, G'$ be two graphs for which there exists $F' \in \mathcal{F}$ such that $\hom(F', G) \neq \hom(F', G')$, and let $\tau, \tau'$ be the corresponding vector representations. If $|V(G)| \neq |V(G')|$ and $G'$ is a blowup of $G$ or vice-versa, simply append the node counts to $\tau, \tau'$ to get $(\tau, |V(G)|) \neq (\tau', |V(G')|)$. If $|V(G)| = |V(G')|$, then $\hom(F', G) \neq \hom(F', G')$ implies that $t(F', G) \neq t(F', G')$. As $\mathcal{D}$ has full support on $\mathcal{F}$, then $\Pr(F = F') > 0$ and therefore $\Pr(F = F') t(F', G) \neq \Pr(F = F') t(F', G')$, which implies $\tau \neq \tau'$. This shows that $\tau$ is $\mathcal{F}$-expressive. If $\mathcal{F} = \mathcal{G}$, then $\tau \neq \tau'$ for any two $G \not\simeq G'$, with analogous argument. Therefore, $\tau$ is in this case complete. $\square$

**Theorem 4.2.** *Let $\mathcal{D}$ be a distribution on $\mathcal{F} \subseteq \mathcal{G}$ with full support. Let $G \in \mathcal{G}$, $\boldsymbol{F} \sim \mathcal{D}^d$, and $\theta \in [0, 1]$. For large enough $d$, $\tilde{\boldsymbol{t}}(\boldsymbol{F}, G)$ is $\mathcal{F}$-expressive with probability at least $1 - \theta$. If $\mathcal{F} = \mathcal{G}$, then, for large enough $d$, $\tilde{\boldsymbol{t}}(\boldsymbol{F}, G)$ is complete with probability at least $1 - \theta$.*

*Proof.* Let $G, G'$ be any two graphs for which there exists $F' \in \mathcal{F}$ such that $\hom(F', G) \neq \hom(F', G')$. First, we consider the noise-free homomorphism density vectors and want to show that

$$\boldsymbol{t}(\boldsymbol{F}, G) = (t(F_1, G), \ldots, t(F_d, G)) \neq (t(F_1, G'), \ldots, t(F_d, G')) = \boldsymbol{t}(\boldsymbol{F}, G') \tag{16}$$

with probability at least $1 - \theta$, where $F_1, \ldots, F_d \sim \mathcal{D}$ iid. To show this, we adapt the proof of Lemma 3 by Welke et al. (2023). Since $t(F, G)$ is $\mathcal{F}$-expressive for $F \sim \mathcal{D}$, then $\mathbb{E}_F[t(F, G)] \neq \mathbb{E}_F[t(F, G')]$. In particular, there exists a set $\mathfrak{F}_{G,G'}$ of outcomes of $F$ with $\Pr(F \in \mathfrak{F}_{G,G'}) = p > 0$ such that for all $F^* \in \mathfrak{F}_{G,G'}$ it holds that $t(F^*, G) \neq t(F^*, G')$. We want that $\Pr[\exists\, i \in \{1, \ldots, d\} : F_i \in \mathfrak{F}_{G,G'}] \geq 1 - \theta$, and thus it must hold that $1 - (1-p)^d \geq 1 - \theta$. Solving for $d$, we obtain that if $d \geq \lceil \frac{\ln(1/\theta)}{\ln\left(\frac{1}{1-p}\right)} \rceil$, then $\boldsymbol{t}(\boldsymbol{F}, G)$ is $\mathcal{F}$-expressive with probability at least $1 - \theta$.

Considering now $\tilde{\boldsymbol{t}}(\boldsymbol{F}, G)$, note that if $t(F^*, G) \neq t(F^*, G')$, then, for any variance $\sigma^2$, it also holds that $\tilde{t}(F^*, G) = t(F^*, G) + \mathcal{N}(0, \sigma^2) \neq t(F^*, G') + \mathcal{N}(0, \sigma^2) = \tilde{t}(F^*, G')$ with probability 1. That is, the patterns for which the noise-free homomorphism densities will distinguish $G$ and $G'$, also work with additive noise. Therefore, $\tilde{\boldsymbol{t}}(\boldsymbol{F}, G)$ is $\mathcal{F}$-expressive with probability at least $1 - \theta$.

If $\mathcal{F} = \mathcal{G}$, then $\tilde{\boldsymbol{t}}(\boldsymbol{F}, G) \neq \tilde{\boldsymbol{t}}(\boldsymbol{F}, G')$ for any two $G \not\simeq G'$, with analogous argument. Therefore, $\tilde{\boldsymbol{t}}(\boldsymbol{F}, G)$ is in this case complete with probability at least $1 - \theta$. $\square$

## B.2 Homomorphism-distinguishing closed graph classes

In Table 1, we report homomorphism-distinguishing closed graph classes for known GNN architectures (Zhang et al., 2024a). For $r$-$\ell$MPNNs, we upper bound the number of edges by the maximum number of edges in outerplanar graphs since fan-cactus graphs are outerplanar (Paolino et al., 2024). For $k$-FGNNs, we can upper bound the number of edges for graphs of bounded treewidth $k$ by considering the number of edges in a $k$-tree, as formalized in the following proposition.

**Proposition B.2.** *Let $\mathcal{F} = \{F : \operatorname{tw}(F) \leq k\}$. Then, any $F \in \mathcal{F}$ with $|V(F)| = m$ nodes has at most $km - \frac{1}{2}k(k+1)$ edges.*

*Proof.* A $k$-tree is an edge-maximal graph of treewidth $k$ and can be constructed by starting with a $(k+1)$-clique and adding new nodes such that each new node is connected to exactly $k$ existing nodes. The initial $(k+1)$-clique has $\frac{1}{2}k(k+1)$ edges. We add $m-(k+1)$ new nodes, where each new node is connected to exactly $k$ existing nodes, thus introducing $k(m-(k+1))$ new edges. Thus, any $F \in \mathcal{F}$ has at most $km - \frac{1}{2}k(k+1)$ edges. $\square$

*Remark* B.3. Maximal outerplanar graphs are 2-trees. Indeed, if we set $k = 2$, we recover our upper bound on the number of edges for outerplanar graphs.

**Proposition B.4.** *Let $\mathcal{F} = \{F : \exists U \subset V(F) \text{ such that } |U| \leq k \text{ and } F \setminus U \text{ is a forest}\}$. Then, any $F \in \mathcal{F}$ with $|V(F)| = m$ has at most $m(k+1) - 1 - \frac{1}{2}(k^2 + 3k)$ edges.*

*Proof.* $F \setminus U$ is a forest and has thus at most $m - k - 1$ edges. Let $F[U]$ denote the subgraph induced by vertex set $U$. $F[U]$ has at most $\frac{1}{2}(k(k-1))$ edges. Every node in $F[U]$ is connected to at most every node in $F \setminus U$. Thus, any $F \in \mathcal{F}$ has at most $m - k - 1 + \frac{1}{2}(k(k-1)) + k(m-k) = m(k+1) - 1 - \frac{1}{2}(3k + k^2)$ many edges. $\square$

**Proposition B.5.** *Let $\mathcal{F}$ denote the class of parallel trees as defined in Gai et al. (2025). Then, any $F \in \mathcal{F}$ with $|V(F)| = m$ nodes has at most $2m - 3$ edges.*

*Proof.* Parallel trees, as defined in Gai et al. (2025), can be obtained by considering a tree $T$ and replacing any edge $uv \in E(T)$ with *parallel edges*, that is, simple paths that share the endpoints $\{u, v\}$. Assume that $T$ has $n_l$ leaves, and denote by $F$ a parallel tree obtained from it. If $T = (\{u, v\}, \{uv\}) = P_2$, then $F$ is a parallel edge which is a series–parallel graph. Therefore, if $n_l \in \{1, 2\}$, i.e., $T$ is a path, $F$ is a series–parallel graph as it is obtained via series composition of series–parallel graphs; then $e(F) \leq 2m - 3$. The bound can be matched by picking $T = P_2$ and adding $m - 2$ nodes, each of them with an edge to $u$ and one to $v$. If $n_l > 2$, then pick a leaf $l_1$ and add edges $\{l_1 l_2, \ldots, l_1 l_{n_l}\}$ from $l_1$ to each of the other leaves. The resulting graph is a series–parallel graph with $m$ nodes. To show this, let $j \in \{1, \ldots, n_l\}$ and consider the $n_l$ root–leaf paths $P_{(j)}$ of $T$, and each of the $n_l$ subgraphs $F_{(j)}$ of $F$ that have been obtained by replacing edges in this path with parallel edges. Each of the $F_{(j)}$ is a series–parallel graph. With the added edges, $l_1$ is now a sink of $F' = (V(F), E(F) \cup \{l_1 l_2, \ldots, l_1 l_{n_l}\})$. With the root as the source, $F'$ is a series–parallel graph. The resulting graph has therefore at most $2m - 3$ edges. $\square$

## B.3 Privacy

**Corollary B.1.** *For any two neighboring graphs $G \sim_1 G'$ with $n$ nodes and for any pattern $F$ it holds that*

$$|t(F, G) - t(F, G')| \leq e(F)d_\square(G, G') = \frac{2e(F)}{n^2}. \qquad (17)$$

*Proof.* We consider $d_\square(G, G') = \delta_\square(G, G')$ as discussed in Appendix A.1. The proof then follows from Lemma 3.1 and Definition 3.2 by direct computation, with the reminder that $e_G(S, S) = 2e(S)$ for any $S \subseteq V(G)$. $\square$

**Proposition 4.4.** *Let $S_{t,*}(G) = \|S_{t,F_1}(G), \ldots, S_{t,F_d}(G)\|_2$ and $\beta > 0$. Let*

$$S_t(G) = \max_{H \in \mathcal{G}} \left( e^{-\beta d_{edge}(G,H)} \max_{H' \in \mathcal{G}: d_{edge}(H,H') \leq 1} \|\boldsymbol{t}(\boldsymbol{F}, H) - \boldsymbol{t}(\boldsymbol{F}, H')\|_2 \right) \qquad (2)$$

*be the $\beta$-smooth sensitivity of $\boldsymbol{t}(\boldsymbol{F}, G)$ at $G$. Then, it holds that $S_{t,*}(G) \geq S_t(G)$.*

*Proof.* Let $\boldsymbol{a}(H)$ be the vector with entries $a_i(H)$ defined by

$$\boldsymbol{a}(H) = (a_1(H), \ldots, a_d(H)), \qquad a_i(H) = \max_{H'': d_{edge}(H,H'') \leq 1} |t(F_i, H) - t(F_i, H'')|. \qquad (18)$$

For any $H'$ with $d_{edge}(H, H') \leq 1$,

$$\|\boldsymbol{t}(\boldsymbol{F}, H) - \boldsymbol{t}(\boldsymbol{F}, H')\|_2 \leq \|\boldsymbol{a}(H)\|_2. \qquad (19)$$

Thus, it holds that

$$S_t(G) = \max_{H \in \mathcal{G}} e^{-\beta d_{\text{edge}}(G,H)} \max_{H': d_{\text{edge}}(H,H') \leq 1} \|\boldsymbol{t}(\boldsymbol{F}, H) - \boldsymbol{t}(\boldsymbol{F}, H')\|_2 \tag{20}$$

$$\leq \max_{H \in \mathcal{G}} e^{-\beta d_{\text{edge}}(G,H)} \|\boldsymbol{a}(H)\|_2 \tag{21}$$

$$\leq \left\| \max_{H \in \mathcal{G}} e^{-\beta d_{\text{edge}}(G,H)} \boldsymbol{a}(H) \right\|_2 \tag{22}$$

$$= \|S_{t,F_1}(G), \ldots, S_{t,F_d}(G)\|_2 = S_{t,*}(G), \tag{23}$$

which concludes the proof. $\qquad\square$

**Theorem 4.5** (Sensitivity of homomorphism density for bounded degree graphs). *Let $G \sim_1 G'$ be two neighboring graphs with $n$ nodes and maximum degree $\Delta_{\max}$. For any pattern $F$ with $m > 1$ nodes, it holds that*

$$|t(F, G) - t(F, G')| \leq \frac{2e(F)}{n^2} \left(\frac{\Delta_{\max}}{n}\right)^{m-2}. \tag{3}$$

*Proof.* Without loss of generality, let $uv \in E(G)$ and $uv \notin E(G')$. We can explicitly compute an upper bound on $|t(F, G) - t(F, G')|$ by counting how many homomorphisms involve $uv$. Note that we do not need to consider homomorphisms that *do not* involve $uv$ as their count is equal for both $G$ and $G'$. First, we can pick any edge of $F$ and map it onto $uv$. For this first step, we have a total of $2e(F)$ choices, as we take into account either order of the endpoints of each edge of $F$. We now map the remaining $m - 2$ nodes of $F$. A third node of $F$ can now be mapped in a total of at most $\Delta_{\max}$ ways, as at most $\Delta_{\max}$ nodes are adjacent to either $u$ or $v$. We can proceed similarly with the remaining nodes. After the first two nodes of $F$ have been mapped, there are then a total of $(\Delta_{\max})^{m-2}$ ways to map the remaining $m - 2$ nodes of $F$. In total, there are therefore at most $2e(F)(\Delta_{\max})^{m-2}$ counts which differ for $G$ and $G'$. Taking the normalization into account, we get $|t(F, G) - t(F, G')| \leq \frac{2e(F)(\Delta_{\max})^{m-2}}{n^m} = \frac{2e(F)}{n^2}\left(\frac{\Delta_{\max}}{n}\right)^{m-2}.$ $\qquad\square$

For the next theorem (Theorem B.6), we first require the following lemma.

**Lemma B.1.** *Consider two multivariate Gaussian distributions $\mathcal{N}(\boldsymbol{\mu}_0, \boldsymbol{\Sigma}_0)$ and $\mathcal{N}(\boldsymbol{\mu}_1, \boldsymbol{\Sigma}_1)$, where $\boldsymbol{\Sigma}_0 = \sigma^2 I_d$ and $\boldsymbol{\Sigma}_1 = e^s \sigma^2 I_d$. Then, if $\alpha\boldsymbol{\Sigma}_0^{-1} + (1-\alpha)\boldsymbol{\Sigma}_1^{-1}$ is positive definite,*

$$D_\alpha(\mathcal{N}(\boldsymbol{\mu}_0, \boldsymbol{\Sigma}_0) \| \mathcal{N}(\boldsymbol{\mu}_1, \boldsymbol{\Sigma}_1)) \tag{24}$$

$$= \frac{\alpha \|\boldsymbol{\mu}_0 - \boldsymbol{\mu}_1\|_2^2}{2[\alpha e^s + (1-\alpha)]\sigma^2} - \frac{d}{2(\alpha-1)}[\alpha s - \ln(\alpha e^s + 1 - \alpha)] \tag{25}$$

*Proof.* Let, for shortness, $(\boldsymbol{\Sigma}_\alpha)^* = \alpha\boldsymbol{\Sigma}_1 + (1-\alpha)\boldsymbol{\Sigma}_0$. From Gil et al. (2013, Table 2), it holds that

$$D_\alpha(\mathcal{N}(\boldsymbol{\mu}_0, \boldsymbol{\Sigma}_0) \| \mathcal{N}(\boldsymbol{\mu}_1, \boldsymbol{\Sigma}_1)) \tag{26}$$

$$= \underbrace{\frac{\alpha}{2}(\boldsymbol{\mu}_0 - \boldsymbol{\mu}_1)^\intercal [(\boldsymbol{\Sigma}_\alpha)^*]^{-1} (\boldsymbol{\mu}_0 - \boldsymbol{\mu}_1)}_{(\star)} - \underbrace{\frac{1}{2(\alpha-1)} \ln \frac{\det(\boldsymbol{\Sigma}_\alpha)^*}{(\det \boldsymbol{\Sigma}_0)^{1-\alpha}(\det \boldsymbol{\Sigma}_1)^\alpha}}_{(\star\star)}. \tag{27}$$

Note that $(\boldsymbol{\Sigma}_\alpha)^* = [\alpha e^s + (1-\alpha)]\sigma^2 I_d$, and therefore

$$(\star) = \frac{\alpha \|\boldsymbol{\mu}_0 - \boldsymbol{\mu}_1\|_2^2}{2[\alpha e^s + (1-\alpha)]\sigma^2} \quad \text{and} \tag{28}$$

$$(\star\star) = -\frac{1}{2(\alpha-1)} \ln \frac{[\alpha e^s + (1-\alpha)]^d \sigma^{2d}}{(\sigma^{2d})^{1-\alpha} e^{sd\alpha}(\sigma^{2d})^\alpha} = -\frac{1}{2(\alpha-1)} \ln \frac{[\alpha e^s + (1-\alpha)]^d}{e^{sd\alpha}} \tag{29}$$

$$= \frac{d}{2(\alpha-1)}[\alpha s - \ln(\alpha e^s + 1 - \alpha)], \tag{30}$$

which concludes the derivation. $\qquad\square$

As our embeddings are in $\mathbb{R}^d$, we need to derive a $d$-dimensional version of Theorem 3.10 for the proof of Theorem 4.6.

**Theorem B.6** (tCDP with Gaussian noise in $\mathbb{R}^d$). *Let $f : \mathcal{X} \to \mathbb{R}^d$ and $g : \mathcal{X} \to \mathbb{R}$ satisfy, for every pair of neighboring databases $x, x' \in \mathcal{X}$ and for $\Delta_f, \Delta_g \geq 0$,*

$$\|f(x) - f(x')\|_2 \leq \Delta_f e^{g(x)/2}, \quad |g(x) - g(x')| \leq \Delta_g. \tag{31}$$

*Let $\mathcal{M} : \mathcal{X} \to \mathbb{R}^d$ be the randomized mechanism defined as $\mathcal{M}(x) = f(x) + \mathcal{N}\left(0, e^{g(x)} I_d\right)$. Then, $\mathcal{M}$ satisfies $\left(\Delta_f^2 + d \cdot \Delta_g^2, \frac{1}{2\Delta_g}\right)$-tCDP.*

*Proof.* We bound the Rényi divergence of two neighboring databases following Lemma B.1, under the conditions in Theorem B.6. Similarly to Bun et al. (2018), we consider $\alpha, s, \gamma \in \mathbb{R}$ with $\alpha(e^s - 1) + 1 \geq \gamma$. Note first that $s = g(x') - g(x)$, as $\mathbf{\Sigma}_1 = e^{g(x')}I_d = e^{g(x')-g(x)}e^{g(x)}I_d = e^s\mathbf{\Sigma}_0$. Due to the $\Delta_g$-lipschitzness of $g$, $s > -\Delta_g$. We can ensure $\alpha(e^s - 1) + 1 \geq \gamma$ by noting that $e^s - 1 \geq e^{-\Delta_g} - 1 \geq -\Delta_g$. Following Bun et al. (2018), we choose $\gamma = \frac{1}{2}$ and can therefore set $\alpha \leq \frac{1}{2\Delta_g}$ to get $\alpha(e^s - 1) + 1 \geq 1 - \alpha\Delta_g \geq \frac{1}{2} = \gamma$.

The first term in Equation (25) is bounded as

$$\frac{\alpha \|\boldsymbol{\mu}_0 - \boldsymbol{\mu}_1\|_2^2}{2\left[\alpha e^s + (1-\alpha)\right]\sigma^2} \leq \frac{\alpha \|\boldsymbol{\mu}_0 - \boldsymbol{\mu}_1\|_2^2}{2\gamma\sigma^2} \leq \alpha\Delta_f^2. \tag{32}$$

The second term in Equation (25) can be bounded via a Taylor expansion of the function $h(s) = \ln\left[\alpha e^s + (1-\alpha)\right]$. First, compute

$$h(0) = 0, \quad h'(s) = \frac{\alpha e^s}{\alpha e^s + (1-\alpha)}, \quad h'(0) = \alpha, \quad h''(s) = \frac{\alpha(1-\alpha)e^s}{\left[\alpha e^s + (1-\alpha)\right]^2}. \tag{33}$$

As in Bun et al. (2018), for $\alpha > 1$ and $\alpha(e^s - 1) + 1 \geq \gamma$ it holds that $0 \leq h''(s) \leq \frac{\alpha(\alpha-1)}{\gamma^2}$. Considering a Taylor expansion in $s = 0$, $h(s) = \alpha s + \frac{1}{2}h''(\zeta)s^2$ for some $\zeta \in [0, s]$, and so

$$\alpha s - h(s) = -\frac{1}{2}h''(\zeta)s^2 \leq \frac{\alpha(\alpha - 1)s^2}{2\gamma^2}. \tag{34}$$

Thus, for $\gamma = 1/2$ the second term in Equation (25) reduces to

$$\frac{d}{2(\alpha - 1)}\left[\alpha s - h(s)\right] \leq \frac{\alpha d s^2}{4\gamma^2} \leq \alpha d \Delta_g^2. \tag{35}$$

Equation (32) and Equation (35) together complete the proof. □

**Theorem 4.6.** *Let $\boldsymbol{t}(\boldsymbol{F}, G)$ be the homomorphism density vector for graph $G$ and pattern set $\boldsymbol{F}$ with $|\boldsymbol{F}| = d$, $\rho' > 0$, and let $S_{t,*}(G)$ be a $\beta$-smooth upper bound to the local sensitivity as per Proposition 4.4. Then, the mechanism*

$$\tilde{\boldsymbol{t}}(\boldsymbol{F}, G) = \boldsymbol{t}(\boldsymbol{F}, G) + \mathcal{N}\left(\mathbf{0}, \frac{[S_{t,*}(G)]^2}{2\rho'} I_d\right) \tag{4}$$

*is $\left(2\rho' + d \cdot 4\beta^2, \frac{1}{4\beta}\right)$-tCDP for neighboring graphs as per Definition 4.3.*

*Proof.* Following the notation in Theorem B.6, let $e^{g(G)} = \frac{[S_{t,*}(G)]^2}{2\rho'}$ and thus $g(G) = \ln\left(\frac{[S_{t,*}(G)]^2}{2\rho'}\right) = 2\ln(S_{t,*}(G)) - \ln(2\rho')$. Therefore, for two adjacent graphs $G \sim G'$, $\Delta_g = |g(G) - g(G')| = 2|\ln S_{t,*}(G) - \ln S_{t,*}(G')| \leq 2\beta$ as $S_{t,*}$ is $\beta$-smooth (Definition A.5). Setting $\|\boldsymbol{t}(\boldsymbol{F}, G) - \boldsymbol{t}(\boldsymbol{F}, G')\|_2 \leq S_{t,*}(G) = \Delta_f e^{g(G)/2} = \Delta_f \left(\frac{[S_{t,*}(G)]^2}{2\rho'}\right)^{1/2} = \Delta_f \frac{S_{t,*}(G)}{\sqrt{2\rho'}}$, it follows that $\Delta_f = \sqrt{2\rho'}$.

From Theorem B.6, $\tilde{\boldsymbol{t}}(\boldsymbol{F}, G)$ is thus $\left(2\rho' + d \cdot 4\beta^2, \frac{1}{4\beta}\right)$-tCDP. □

## B.4 EXPRESSIVE AND PRIVATE GRAPH REPRESENTATIONS

**Theorem 4.8.** *Let $\mathcal{D}$ be a distribution on $\mathcal{F} \subseteq \mathcal{G}$ with full support. Let $G \in \mathcal{G}$ be a graph and $\boldsymbol{F} = (F_1, \ldots, F_d) \sim \mathcal{D}^d$ be a vector of patterns. Then, the graph representation $\tilde{\boldsymbol{t}}(\boldsymbol{F}, G) = \boldsymbol{t}(\boldsymbol{F}, G) + \mathcal{N}\left(\mathbf{0}, \frac{[S_{t,*}(G)]^2}{2\rho'}I_d\right)$ is $\mathcal{F}$-expectation-expressive and $(2\rho' + d \cdot 4\beta^2, \frac{1}{4\beta})$-tCDP, where $\rho' > 0$ and $S_{t,*}(G)$ is a $\beta$-smooth upper-bound on the local sensitivity of $\boldsymbol{t}(\boldsymbol{F}, G)$. If $\mathcal{F} = \mathcal{G}$, then $\tilde{\boldsymbol{t}}(\boldsymbol{F}, G)$ is also expectation-complete.*

*Proof.* From Theorem 4.6, $\tilde{\boldsymbol{t}}(\boldsymbol{F}, G)$ is $(2\rho' + d \cdot 4\beta^2, \frac{1}{4\beta})$-tCDP. From Theorem 4.1, $\tilde{\boldsymbol{t}}(\boldsymbol{F}, G)$ is $\mathcal{F}$-expectation-expressive and expectation-complete if $\mathcal{F} = \mathcal{G}$. $\qquad\square$

**Proposition 4.9.** *Consider a graph $G$ with $n$ nodes and let $\mathcal{F}$ be a class of patterns. For a chosen privacy parameter $\rho' > 0$, the Gaussian noise necessary to obtain a specific privacy guarantee in Theorem 4.8 has variance $\sigma^2 = \mathcal{O}\left((\max_{F \in \mathcal{F}} e(F))^2 / n^4\right)$.*

*Proof.* From Theorem 4.5, the local sensitivity of each pattern is $\mathcal{O}(e(F)/n^2)$. The vector-wise smooth sensitivity in Proposition 4.4 is not smaller than the largest local sensitivity and therefore $S_{t,*}(G) = \mathcal{O}\left(\max_{F \in \mathcal{F}} e(F)/n^2\right)$. For a fixed $\rho'$, the variance of the noise in Theorem 4.6 is $\sigma^2 = \mathcal{O}\left((\max_{F \in \mathcal{F}} e(F))^2 / n^4\right)$. $\qquad\square$

## C  TECHNICAL DETAILS

### C.1  EXPECTED BEHAVIOR OF AUC UNDER GAUSSIAN NOISE

For a subset of our experiments, we can describe the expected behavior of the AUC for increasing amounts of additive Gaussian noise as follows.

**Proposition C.1.** *In a binary classification setting with separable classes, the AUC curve follows the error function* erf *for embeddings perturbed with additive Gaussian noise.*

*Proof.* In a binary classification setting, let $C_0$ and $C_1$ be the two classes with means $\mu_0$ and $\mu_1$. Assume a one-dimensional setting and that the classes are separated by $\mu_1 - \mu_0 = \triangle > 0$. If the points in each class are perturbed by additive noise $\mathcal{N}(0, \sigma^2)$, the distance $B$ between points from the two classes is $B \sim \mathcal{N}(\triangle, 2\sigma^2)$. With these assumptions, the AUC is the probability that points are not misranked and thus $\text{AUC} = \Pr[B > 0] = \Pr[\mathcal{N}(\triangle, 2\sigma^2) > 0] = \Phi(\frac{\triangle}{\sigma\sqrt{2}}) = \frac{1}{2}\left[1 + \text{erf}\left(\frac{\triangle}{2\sigma}\right)\right]$, where $\Phi$ is the Gaussian cumulative density function. $\qquad\square$

As the private mechanism we rely on uses additive Gaussian noise, Proposition C.1 applies. In a practical setting, even though we may not have perfectly separated classes, we thus expect the AUC curve to roughly follow the erf function for increasing amounts of noise.

### C.2  STOCHASTIC BLOCK MODEL

To highlight how different pattern classes can heavily influence classification performance, we use a simple two-block stochastic block model (SBM) to generate a dataset where certain pattern classes are informative while others are not. In the SBM, graphs are sampled according to a fixed, class-independent mean edge probability $q \in [0, 1]$ and a class parameter $\zeta_c$, which controls the bias towards same-block edges. In this dataset, up to a $\mathcal{O}(1/n)$ factor, tree densities are unaffected by $\zeta_c$ and do not discriminate between classes. Instead, cycle densities depend on $\zeta_c^m$ for a cycle with $m$ edges and are thus able to effectively distinguish between classes. We thus expect cycles to perform significantly better than trees on this dataset, as for trees the class signal is carried only by a term that scales with $1/n$. In fact, in the large graph limit, the result holds with no $\mathcal{O}(1/n)$ term: for graphons on this SBM dataset, cycles can discriminate between classes while trees cannot (see Lemma C.2).

**Lemma C.1** (Homomorphism densities for SBM). *Consider a graph $G \in \mathcal{G}$ sampled from the stochastic block model (SBM) on $n$ nodes defined as follows. To define the blocks, draw labels*

$\beta(v) \in \{1, -1\}$ *iid with probabilities* $\Pr[\beta(v) = 1] = \Pr[\beta(v) = -1] = 1/2$. *The probability of an edge on distinct, unordered pairs of nodes* $\{u, v\} \in V(G)$ *is defined as*

$$\Pr[uv \in E(G) \mid \beta] = q + \zeta_c \beta(u)\beta(v) \quad (\text{with } u \neq v), \tag{36}$$

*where* $q \in [0, 1]$ *and* $|\zeta_c| \leq \min(q, 1-q)$. *We consider the class of* $G$ *to be determined by the value of* $\zeta_c$. *For any pattern* $F \in \mathcal{F} \subseteq \mathcal{G}$ *with* $e(F)$ *edges and* $m = |V(F)|$ *nodes it holds that*

$$\mathbb{E}[t(F, G)] = \sum_{\substack{S \subseteq E(F) \\ \Delta_S(w) \text{ is even } \forall w \in V(F)}} q^{e(F) - |S|} \zeta_c^{|S|} + \mathcal{O}\left(\frac{1}{n}\right), \tag{37}$$

*where* $\Delta_S(w)$ *denotes the number of edges in* $S$ *that are incident to* $w$. *In particular, if* $T$ *is a tree it holds that* $\mathbb{E}[t(T, G)] = q^{e(T)} + \mathcal{O}(1/n)$, *and if* $C_m$ *is a cycle with* $m$ *edges it holds that* $\mathbb{E}[t(C_m, G)] = q^m + \zeta_c^m + \mathcal{O}(1/n)$, *where expectations are taken over the SBM sampling.*

*Proof.* Let $\psi : V(F) \to V(G)$. Then, the homomorphism density $t(F, G)$ can be written as

$$t(F, G) = \frac{1}{n^m} \sum_\psi Z_\psi \quad \text{where} \quad Z_\psi = \prod_{ab \in E(F)} \mathbf{1}_{\psi(a)\psi(b) \in E(G)}. \tag{38}$$

The probability of an edge in $G$ is a function of the random variable $\beta$. Thus, using the linearity of expectation and the law of total expectation we can write the expected homomorphism density as $\mathbb{E}[t(F, G)] = \frac{1}{n^m} \sum_\psi \mathbb{E}[Z_\psi] = \frac{1}{n^m} \sum_\psi \mathbb{E}_\beta[\mathbb{E}[Z_\psi \mid \beta]]$.

**Case 1:** If $\psi$ is injective on nodes, then for every edge $ab \in E(F)$ the nodes $a$ and $b$ are mapped to a distinct unordered pair $\{\psi(a), \psi(b)\} \in V(G)$. For all pairs of nodes of $G$ the probabilities that $\psi(a)\psi(b) \in E(G)$ are then conditionally independent and can be factorized as follows.

$$\mathbb{E}[Z_\psi \mid \beta] = \prod_{ab \in E(F)} \Pr[\psi(a)\psi(b) \in E(G) \mid \beta] = \prod_{ab \in E(F)} q + \zeta_c \beta(\psi(a))\beta(\psi(b)). \tag{39}$$

By the distributive property, we can rewrite

$$\mathbb{E}[Z_\psi \mid \beta] = \prod_{ab \in E(F)} q + \zeta_c \beta(\psi(a))\beta(\psi(b)) = \sum_{S \subseteq E(F)} q^{e(F) - |S|} \zeta_c^{|S|} \prod_{ab \in S} \beta(\psi(a))\beta(\psi(b)). \tag{40}$$

For each node $w \in V(F)$ the term $\prod_{ab \in S} \beta(\psi(a))\beta(\psi(b))$ appears exactly $\Delta_S(w)$ times in each summand. It then holds that $\prod_{ab \in S} \beta(\psi(a))\beta(\psi(b)) = \prod_{w \in V(F)} \beta(\psi(w))^{\Delta_S(w)}$. As $\psi$ is injective on nodes, the random variables $\beta(\psi(w))$ are independent and the expectation over $\beta$ can be factorized as

$$\mathbb{E}_\beta[\mathbb{E}[Z_\psi \mid \beta]] = \mathbb{E}_\beta\left[\sum_{S \subseteq E(F)} q^{e(F) - |S|} \zeta_c^{|S|} \prod_{w \in V(F)} \beta(\psi(w))^{\Delta_S(w)}\right] \tag{41}$$

$$= \sum_{S \subseteq E(F)} q^{e(F) - |S|} \zeta_c^{|S|} \prod_{w \in V(F)} \mathbb{E}_\beta\left[\beta(\psi(w))^{\Delta_S(w)}\right]. \tag{42}$$

For any $\beta(\psi(w))$ it holds that $\mathbb{E}_\beta[\beta(\psi(w))^{\Delta_S(w)}]$ is equal to 0 if $\Delta_S(w)$ is odd, and to 1 otherwise. Therefore, each term in the sum is 1 if and only if $\Delta_S(w)$ is even for every node in the graph $(V(F), S)$. If $\psi$ is injective on nodes, it therefore holds that

$$\mathbb{E}[Z_\psi] = \sum_{\substack{S \subseteq E(F) \\ \Delta_S(w) \text{ is even } \forall w \in V(F)}} q^{e(F) - |S|} \zeta_c^{|S|}. \tag{43}$$

**Case 2:** If $\psi$ is not injective, some edges of $F$ are mapped to the same unordered pair in $G$. Therefore, the conditional independence necessary to obtain Equation (39) does not hold. In this case, we need to consider the image graph $H_\psi$ with node set $|\psi(V(F))| < |V(F)|$ and edge set $E(H_\psi)$ induced by the set of distinct, unordered pairs to which $\psi$ maps to. A similar derivation as above shows that in this case

$$\mathbb{E}[Z_\psi] = \sum_{\substack{S \subseteq E(H_\psi) \\ \Delta_S(w) \text{ is even } \forall w \in V(H_\psi)}} q^{e(H_\psi) - |S|} \zeta_c^{|S|}. \tag{44}$$

Of the possible $n^m$ mappings $\psi$, there are $n(n-1)\cdots(n-m+1)$ injective mappings. After normalization by $1/n^m$, there is therefore at most a fraction of $1 - \frac{n(n-1)\cdots(n-m+1)}{n^m} = \binom{m}{2}/n + \mathcal{O}(1/n^2) = \mathcal{O}(1/n)$ non-injective maps. Summing over all $\psi$ leads then to the stated expected value for the homomorphism density.

The results for cycles and trees can be obtained by noting that, except for the empty set, a cycle with $m$ nodes has a single subset where every node has even degree (itself), while a tree has no other subsets where every node has even degree. $\square$

**Lemma C.2** (Homomorphism densities for SBM on graphons). *Consider a two-block graphon defined in accordance to the SBM setting in Lemma C.1. Let therefore $W$ be a graphon $W : [0,1]^2 \to [0,1]$ defined as $W(x,y) = q + \zeta_c s(x)s(y)$, with $s = \mathbf{1}_{[0,1/2]} - \mathbf{1}_{(1/2,1]}$. For a cycle $C_m$ with $m$ edges it holds that $t(C_m, W) = q^m + \zeta_c^m$. For a tree $T$ it holds that $t(T, W) = q^{e(T)}$.*

*Proof.* Consider the operator associated to the graphon $\mathcal{T}_W : L^2[0,1] \to L^2[0,1]$ defined as $(\mathcal{T}_W f)(x) = \int_0^1 W(x,y)f(y)dy$. From Lovász (2012, Section 7.5), this is a Hilbert-Schmidt operator and thus has a discrete spectrum. By direct computation, we obtain that $\mathcal{T}_W f = q\langle \mathbf{1}, f\rangle \mathbf{1} + \zeta_c\langle s, f\rangle s$, where $\mathbf{1}$ denotes the constant function $x \mapsto 1$ and $\langle \cdot, \cdot \rangle$ is the inner product on $L^2[0,1]$. Thus the range of the operator is spanned by $\{\mathbf{1}, s\}$; $\mathcal{T}_W$ has rank at most 2 and thus at most two non-zero eigenvalues. The eigenfunctions of $\mathcal{T}_W$ are $\mathbf{1}$ and $s$ with eigenvalues $q$ and $\zeta_c$, respectively. As the homomorphism density for a cycle $C_m$ is $t(C_m, W) = \sum_k \lambda_k^m$, where $\lambda_k$ is the $k$-th eigenvalue of the graphon operator (Lovász, 2012, Equation 7.22), we get $t(C_m, W) = q^m + \zeta_c^m$. For the result for trees, we proceed by induction. Given a tree $T$, consider a leaf node $\ell$ with a neighbor $u$, and denote the leaf and neighbor variables with $x_\ell$ and $x_u$. The homomorphism density of $T$ in $W$ can be computed as $t(T, W) = \int_{[0,1]^{|V(T)|}} \prod_{ab \in E(T)} W(x_a, x_b) \prod_{v \in V(T)} dx_v$ (Lovász, 2012, Section 7.2). We can rewrite this as

$$t(T, W) = \int_{[0,1]^{|V(T)\setminus\{\ell\}|}} \left( \int_0^1 W(x_\ell, x_u)dx_\ell \right) \prod_{ab \in E(T)\setminus \ell u} W(x_a, x_b) \prod_{v \in V(T)\setminus\{\ell\}} dx_v. \quad (45)$$

By direct computation, we obtain $\int_0^1 W(x_\ell, x_u)dx_\ell = \int_0^1 (q + \zeta_c s(x_\ell)s(x_u))dx_\ell = q + \zeta_c s(x_u) \int_0^1 s(x_\ell)dx_\ell = q$, since $\int_0^1 s(x)dx = 0$. Therefore, $t(T, W) = qt(T \setminus \{\ell\}, W)$ and, taking the induction step and integrating over the remaining nodes gives the result for trees, $t(T, W) = q^{e(T)}$. $\square$

**Proposition C.2** (Homomorphism counts for cycles, Lovász 1967, Example 5.11). *For a cycle $C_m$ on $m$ nodes, $\hom(C_m, G)$ is the trace of the $m$-th power of the adjacency matrix of $G$, and therefore $\hom(C_m, G) = \sum_{i=1}^n \lambda_i^m$, where $\lambda_1, \ldots, \lambda_n$ are the eigenvalues of the adjacency matrix of $G$.*

# D EXPERIMENTS

In this section, we provide details for our experimental evaluation and present additional results.

## D.1 SETUP AND DETAILS ON EXPERIMENTS

**Setup and hyperparameters.** In our experiments, we consider different privacy budgets by choosing different values of $\rho' \in \{10^{-8}, 10^{-6}, 10^{-5}, 10^{-4}, 10^{-3}, 5 \cdot 10^{-3}, 10^{-2}, 5 \cdot 10^{-2}, 1\}$ and set $\beta = \rho'/5$. We upper bound the smooth sensitivities by evaluating Equation (1) up to $d_{\text{edge}}(G, G') = 6$. For visualization purposes, we convert our tCDP guarantees into $(\epsilon, \delta)$-DP guarantees using Lemma A.2 in Appendix A.2; $(\epsilon, \delta)$-DP guarantees are easier to interpret. We use $\delta = 10^{-6}$ for all our guarantees. This choice respects the standard requirement $\delta \ll 1/e(G)$ (Sajadmanesh et al., 2023) and is a common choice in related literature. Our choice of $\delta$, together with the choice of $\beta$, and the range of values of $\rho'$ we experiment with allow us to obtain meaningful privacy protection and good performance for reasonable privacy budgets. In fact, privacy budgets roughly in the range $\epsilon \in (0, 10]$ are generally understood to provide meaningful privacy protection in graph machine learning (Wu et al., 2022; Sajadmanesh & Gatica-Perez, 2021). If not differently specified, for each dataset we sample three pattern vectors $\mathbf{F}$ of size $d = 50$, with the sampling

strategy described in Welke et al. (2023). For each value of $\rho'$, we perform three runs for each of the sampled pattern vectors with three different seeds, leading to a total of 9 runs. We train our models on the noisy homomorphism density embeddings, and test on unseen, not noisy embeddings.

**Experiments on `OGBG` data.** For the molecular datasets, we take $\Delta_{\max} = 10$ for `MOLHIV`, and $\Delta_{\max} = 6$ for `MOLBACE`, `MOLBBBP`, and `MOLLIPO`. For each dataset we sample patterns of treewidth 1. For our classification tasks, we train on the private homomorphism densities to predict the class of unseen graphs. We consider the 1000 and 100 nearest neighbors in a nearest neighbors classifier for `MOLHIV` and `MOLBACE`, respectively. We consider 200 estimators in a random forest classifier for `MOLBBBP`. We evaluate the performance of our classifiers and report the classification AUC for different privacy budgets. For the regression task on `MOLLIPO`, we use an SVR with linear kernel and default hyperparameters from scikit-learn, except for `epsilon = 0.2`.

**Experiments on network data.** For the network datasets, we take $\Delta_{\max} = n$, as there is no upper bound on the maximum degree that we can infer from domain knowledge. For each dataset we sample patterns of treewidth 1 and append node counts. We train on the private homomorphism densities to predict the class of unseen graphs. We consider the 300 nearest neighbors in a nearest neighbors classifier for `REDDIT-BINARY`. We consider 200 and 50 estimators in a random forest classifier for `REDDIT-MULTI-5K` and `STARGAZERS`, respectively. We evaluate the performance of our classifiers and report the classification AUC and accuracy for different privacy budgets.

**Experiments on synthetic data.** For the `SBM` dataset, we consider graphs with $n = 200$ nodes, and classes defined by $\zeta \in \{0.08, 0.16, 0.24, 0.32\}$ as described in Appendix C.2 to generate 100 graphs per class. We use a Chernoff bound to estimate $\Delta_{\max}$ with probability $p = 0.995$. We use a nearest neighbors classifier and consider the 5 nearest neighbors. For cycle patterns, the homomorphism densities for a graph $G$ can be quickly computed using the eigenvalues of $G$ as in Proposition C.2. To consider a distribution with full support on the cycle graph patterns, one can, e.g., sample the number of nodes $m$ with a Poisson distribution and then consider all cycles with number of nodes up to $m$. In our experiments, we consider cycles up to $m = 10$ nodes for a total of $d = 8$ patterns.

**Treewidth trade-off.** On `MOLHIV` we additionally consider patterns with maximum treewidths of 2 and 3. For these results, we ensure that at least 25% of the patterns match the maximum treewidth.

**Privacy attacks.** To empirically test our privacy guarantees, we consider the following attack scenario. We assume a strong attacker that has access to the vector of patterns $\boldsymbol{F}$ and to the original set of graphs $\{G_1, \ldots, G_N\}$. For each $G_i \in \{G_1, \ldots, G_N\}$, the attacker can compute the true homomorphism density vector $\boldsymbol{t}(\boldsymbol{F}, G_i)$. The attacker has access to the private homomorphism densities and their goal is to recover an unknown graph $G$ from the private $\tilde{\boldsymbol{t}}(\boldsymbol{F}, G)$ by matching it with one of the computed $\boldsymbol{t}(\boldsymbol{F}, G_i)$. Concretely, we train a nearest neighbors classifier on the (noise-free) homomorphism densities and use this classifier to perform the attack. We compute the Top-1 attack accuracy by recording whether the nearest neighbor of $\tilde{\boldsymbol{t}}(\boldsymbol{F}, G)$ is the true graph's density $\boldsymbol{t}(\boldsymbol{F}, G)$, which allows the attacker to identify $G$. We compute the Top-10 attack accuracy by recording whether the true graph appears in the 10 nearest neighbors. This provides an empirical lower bound to the attacker's abilities, but the possibility of a stronger attacker is not excluded.

**Node features.** To evaluate whether the inclusion of node features can be beneficial, we consider aggregated node features which do not consider the structure of the graph (i.e., which are edge-private). More specifically, we consider the following statistics on node features: mean, standard deviation, median, maximum, minimum, and sum. We evaluate both the performance of node features used as embeddings alone, and appended to the private homomorphism densities, to establish whether using the private homomorphism densities with node features leads to performance gains.

**Ablation experiments.** To probe the effectiveness of the homomorphisms density embeddings we conduct two ablation studies. First, to further justify our choice to rely on smooth sensitivities, we consider noise scaled to *global* sensitivities and investigate the performance of the resulting noisy homomorphisms densities on the `OGBG` datasets. Second, we consider different values for the num-

ber of sampled homomorphisms densities $d$, to investigate wether smaller or larger homomorphisms density vectors can provide better performance.

**Comparison with GNN baselines.** We compare our results with common approaches to achieve edge-level DP for graph classification. As a first baseline, we use Randomized Response (RR) (Wang et al., 2016) to perturb the structures of graphs and train a GNN on the perturbed graphs. RR perturbs each entry of the (undirected) adjacency matrix $A$ of a graph as follows: each entry $A_{ij}$ is independently perturbed from a 0 to a 1 (and vice-versa) with probability $1 - p$. That is, RR leaves an entry in the adjacency matrix unchanged with probability $p$, and flips it with probability $1 - p$. If $p = e^\epsilon/(1 + e^\epsilon)$ the resulting perturbed graph is $\epsilon$-edge-DP (Wang et al., 2016). As an additional baseline, we also use the degree-preserving variant of RR (DPRR), recently proposed by Hidano & Murakami (2024). With DPRR, the nodes of the perturbed graphs keep approximately the same degree as the nodes of the unperturbed graphs, which results in better performance, as well as more efficient training (Hidano & Murakami, 2024) when compared to RR. For both the RR and the DPRR baselines, we thus perturb the training graphs for our `OGBG` experiments and test the performance of a GIN (Xu et al., 2019) architecture for $\epsilon = 1$. For these experiments, we rely on the hyperparameters in Welke et al. (2023), not including dropout layers. Note that the notion of $\epsilon$-edge-DP obtained via RR/DPRR does not perfectly coincide with the smooth sensitivity framework we leverage. In addition, our homomorphism density embeddings guarantee expressivity in expectation, while DP GNNs offer no formal expressivity guarantees.

## D.2 ADDITIONAL RESULTS

In this section we present additional experimental results.

**Results for `OGBG` data.** Our experiments on `MOLHIV` and `MOLBACE`, which we display in Figure 3, show that our approach successfully obtains a private embedding which retains discrimination abilities that are comparable to that of a non-private embedding ($\epsilon = \infty$). At the same time, the attacker performance drastically decreases for reasonable values of $\epsilon$, while being close to 1 for $\epsilon = \infty$. The classification AUC follows the error function, empirically confirming the formal connection between privacy and AUC discussed in Proposition C.1. This result is of great practical utility, as it allows to determine the maximum privacy budget for a given desired AUC, and vice-versa the predicted AUC for a given privacy budget.

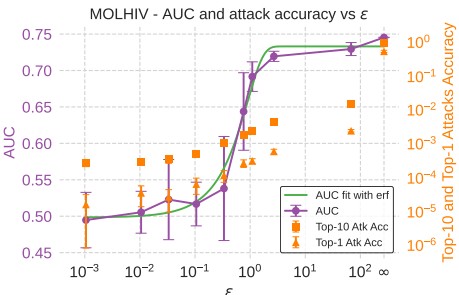 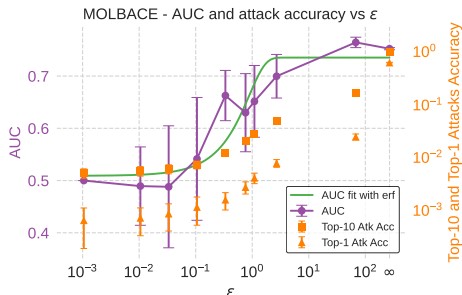

(a) Classification AUC and attack accuracy vs privacy budget on `MOLHIV` with $k$-NN.

(b) Classification AUC and attack accuracy vs privacy budget on `MOLBACE` with $k$-NN.

Figure 3: Visualizations for two of our experiments on `MOLHIV` and `MOLBACE`. We report average results with error bars of 2 standard deviations across 9 runs.

**Node features.** In Table 3, we can see that node features overall achieve reasonable performance. However, combining the node features with our private embeddings with $\epsilon = 1$ provides better performance. Therefore, we can render the homomorphism density embeddings more informative by additionally considering node features.

**Results on network data.** We also perform experiments on the network datasets `REDDIT-BINARY`, `REDDIT-MULTI-5K` (Xiang et al., 2024), and GitHub `STARGAZERS`

Table 3: Utility and attack accuracy for our experiments on `OGBG` datasets. As utility metric, we use the regression RMSE for `MOLLIPO` and the classification AUC for `MOLHIV`, `MOLBBBP`, and `MOLBACE`. We include results where we concatenate node features (NF) to the private homomorphism density embeddings. We report average results and standard deviations across 9 runs.

|  |  | MOLHIV ↑ | MOLBBBP ↑ | MOLBACE ↑ | MOLLIPO ↓ |
|---|---|---|---|---|---|
| Private | Utility | 0.692 (0.020) | 0.602 (0.005) | 0.652 (0.069) | 1.086 (0.004) |
| ($\epsilon = 1$) | Attack | 0.003 (< 0.001) | 0.025 (0.003) | 0.027 (0.002) | 0.011 (0.002) |
| Non private | Utility | 0.745 (< 0.001) | 0.644 (0.008) | 0.752 (0.002) | 1.055 (0.002) |
| ($\epsilon = \infty$) | Attack | 0.955 (0.020) | 1.000 (< 0.001) | 0.990 (< 0.001) | 0.992 (0.006) |
| Private + NF | Utility | 0.731 (< 0.001) | 0.604 (0.005) | 0.739 (0.005) | 1.086 (0.004) |
| ($\epsilon = 1$) | Attack | 0.003 (< 0.001) | 0.025 (0.003) | 0.027 (0.002) | 0.011 (0.002) |
| Non private + NF | Utility | 0.750 (0.003) | 0.644 (0.010) | 0.739 (0.003) | 1.053 (0.002) |
| ($\epsilon = \infty$) | Attack | 0.970 (0.024) | 1.000 (< 0.001) | 0.990 (< 0.001) | 0.992 (0.006) |
| Features only | Utility | 0.721 (< 0.001) | 0.603 (0.002) | 0.730 (< 0.001) | 1.085 (< 0.001) |

Table 4: Utility and attack accuracy for our experiments on network datasets. As utility metric, we use the classification accuracy and the classification AUC for all datasets. We report average results and standard deviations across 9 runs.

|  |  | REDDIT-BINARY | REDDIT-MULTI-5K | STARGAZERS |
|---|---|---|---|---|
| Private | Accuracy | 0.758 (< 0.001) | 0.416 (0.021) | 0.590 (0.015) |
| ($\epsilon = 1$) | AUC | 0.775 (0.009) | 0.749 (0.014) | 0.609 (0.026) |
|  | Attack | 0.046 (0.016) | 0.018 (0.004) | 0.004 (0.001) |
| Non private | Accuracy | 0.771 (0.031) | 0.508 (0.007) | 0.670 (0.003) |
| ($\epsilon = \infty$) | AUC | 0.844 (0.039) | 0.805 (0.002) | 0.729 (0.002) |
|  | Attack | 0.999 (< 0.001) | 1.000 (< 0.001) | 0.959 (< 0.001) |

(Rozemberczki et al., 2020), presented in Table 4. We obtain accuracy and classification AUC comparable to those in Hidano & Murakami (2024, Figure 5), showing that the noisy homomorphism density embeddings also provide good performance on larger network graphs. We remark that our embeddings include node counts and that on `REDDIT-BINARY` and `REDDIT-MULTI-5K` node and edge counts alone achieve good performance (Schulz & Welke, 2019, Figure 1). Compared to Hidano & Murakami (2024), we rely on significantly simpler and less resource-expensive classifiers. We could not reproduce the results in Hidano & Murakami (2024) due to out-of-memory errors, and we thus refer to Hidano & Murakami (2024) for a comparison.

## D.3 ABLATIONS AND COMPARISONS WITH BASELINES

In this section, we perform additional experiments to evaluate the performance of our embeddings for different values of $d$, i.e., the number of homomorphism densities we sample. Next, we compare our results with those obtained considering a global sensitivity notion, to further motivate our choice to rely on the smaller smooth sensitivities.

**Ablation on $d$.** In Table 5, we observe that smaller embeddings, i.e., embeddings which consider fewer patterns, tend to perform worse. In particular, smaller embeddings have a much higher variance, as their performance more heavily depends on having sampled patterns which are informative for the task.

**Ablation with global sensitivity.** In Table 6, we observe that private embeddings obtained by relying on global sensitivity perform significantly worse than the ones obtained relying on local sensitivity.

Table 5: Utility for our experiments on OGBG using private homomorphisms density embeddings with varying sizes for $d$. Results are for $\epsilon = 1$. We report average results and standard deviations across 9 runs.

| $d$ | MOLHIV↑ | MOLBBBP↑ | MOLBACE↑ | MOLLIPO↓ |
|---|---|---|---|---|
| 10 | 0.556 (0.153) | 0.503 (0.070) | 0.552 (0.156) | 1.099 (0.001) |
| 20 | 0.572 (0.129) | 0.547 (0.074) | 0.617 (0.134) | 1.097 (0.001) |
| 30 | 0.540 (0.171) | 0.554 (0.036) | 0.544 (0.098) | 1.099 (0.001) |
| 40 | 0.592 (0.089) | 0.570 (0.067) | 0.498 (0.134) | 1.098 (0.001) |
| 50 | 0.692 (0.020) | 0.602 (0.005) | 0.652 (0.069) | 1.086 (0.001) |

Table 6: Utility for our experiments on OGBG using private homomorphism densities obtained with noise scaled using the *global* sensitivity of the homomorphism densities, compared to that obtained with noise scaled with smooth sensitivity. We report average results and standard deviations across 9 runs for $\epsilon = 1$. **Bold** marks the best results.

| | MOLHIV↑ | MOLBBBP↑ | MOLBACE↑ | MOLLIPO↓ |
|---|---|---|---|---|
| Global sensitivity | 0.492 (0.023) | 0.500 (< 0.001) | 0.520 (0.133) | 1.199 (< 0.001) |
| Smooth sensitivity | **0.692** (0.020) | **0.602** (0.005) | **0.652** (0.069) | **1.086** (0.004) |

## D.4 RUNTIMES

We measured the time to compute homomorphism density embeddings for MOLHIV with increasing maximum treewidth of $tw = \{1, 2, 3\}$ and REDDIT-BINARY with $tw = 1$. The results presented in Table 7 are averaged over 3 seeds. The runtime of the homomorphism density computation for $tw = 1$ for MOLHIV is comparable to training GIN on MOLHIV for 200 epochs with the RR or DPRR baselines. The runtime of our homomorphism density computation is therefore comparable to that of existing methods, showing that our approach is also competitive from a runtime perspective. Once the homomorphism density vectors are computed, the training runtime itself is negligible; the embeddings are informative and provide competitive performance with simple and efficient approaches such as $k$-NN or Random Forest. We also remark that the homomorphism density approach provides expressivity guarantees which are not provided by the RR/DPRR+GIN baselines.

Table 7: Runtimes for the computation of the homomorphism density embeddings $t(\boldsymbol{F}, G)$, and for the training of the GNN baselines for 100 epochs with $\epsilon = 1$. Values reported with a star (*) for RR/DPRR are obtained from Hidano & Murakami (2024). Values are reported in seconds.

| Method | MOLHIV | | | REDDIT-BINARY |
|---|---|---|---|---|
| | $tw = 1$ | $tw \leq 2$ | $tw \leq 3$ | |
| $t(\boldsymbol{F}, G)$ | 2369 (4) | 2432 (52) | 3916 (1088) | 602 (136) |

| | MOLHIV | REDDIT-BINARY |
|---|---|---|
| RR | 1153 (159) | > 800* |
| DPRR | 1214 (101) | > 200* |

