# OpenReview forum: "On the trade-off between expressivity and privacy in graph representation learning"
_ICLR.cc/2026/Conference — ICLR 2026 Poster_

### Official Review · Reviewer_dNzD · 2025-10-30

**Soundness:** 3
**Presentation:** 3
**Contribution:** 2
**Rating:** 6
**Confidence:** 3

**Summary:**

The paper studies the trade-off between privacy and expressivity in graph representation learning. It proposes using homomorphism density vectors as graph embeddings to achieve both expressive power and formal privacy guarantees. The paper derives the trade-off in two parts: firstly, if homomorphism density vectors are used as graph embeddings, then with any noise added to them, in expectation, they will achieve expressiveness. Secondly, it proposed to use a $\beta$-smoothing sensitivity bound and leverage Gaussian noise to achieve tCDP. Extensive theoretical and experimental results are conducted to highlight and illustrate the advantage of using homomorphism density vectors as embeddings.

**Strengths:**

- Provide a strong theoretical guarantee for the expressivity and privacy of graph representation learning.
- The experimental results highlight the validity and soundness of the theoretical claims.
- The proposed method has been proven to be useful through experimental results and is robust against attacks.

**Weaknesses:**

- The privacy guarantee is provided for local sensitivity. Although it is demonstrated that very similar graphs can still be distinguished by the noisy homomorphism density vectors, the privacy guarantee does not meet the DP standard.
- The expressivity guarantee is analyzed in expectation, which is orthogonal to the privacy analysis. Since the DP noise is only added once, the expressivity will be biased to that sampled noise, leading to the degradation of the guarantee.
- Some of the critical theoretical annotations are provided in the Appendix, making it hard to follow the flow of the paper.

**Questions:**

- In line 206, why is the sensitivity of Theorem 3.10 a local sensitivity if the neighboring database is considered across the input domain?
- By the definition of F-expectation-expressivity and Expectation-completeness, what are the $X$ for your analysis and proposed method?
- By expectation w.r.t $X$, the F-expectation-expressivity and Expectation-completeness have to consider all of the possible inputs of the input space? Then, can you elaborate on how to use this expectation in your expressivity guarantee?

---

> ### Author Response · Authors · 2025-11-20
>
> We thank the reviewer for their time and the positive feedback! We believe there might be some misunderstanding on our contribution, which we will address in more detail below.
>
> **Addressing Weakness 1:**
>
> Our guarantees rely on _smooth_ sensitivity [1], which is an upper bound on local sensitivity. Smooth sensitivity is a good choice in our case, as global sensitivity would require too much noise to be added (and not be useful in practice, see the ablation study in Table 5 in our submission) while local sensitivity alone, as you correctly pointed out, would not yield DP. Instead, our choice to use noise calibrated to the smooth sensitivity does provide us with DP guarantees, while reducing the amount of noise added if compared to global sensitivity.  So: our privacy guarantee **does** meet the DP standard.
>
> **Addressing Weakness 2:**
>
> As you correctly point out, the expressivity guarantee is in expectation. This is a necessity, since adding noise (to guarantee DP) makes it impossible to give a deterministic guarantee. However, given that we add noise with mean zero, the expressivity guarantee is actually **unbiased** in expectation and there is no degradation of the guarantee in this sense. This is the key observation that allows us to study expressivity and privacy. Thank you for pointing out that this was not made clear enough in our submission; we will make sure to stress this more in the revised version.
>
> **Addressing Weakness 3:**
>
> Thank you for the feedback. Given that we are able to add another page for the final version, could you give us more detailed feedback on what sections could benefit most from some additional information?
>
> **Answer to Question 1:**
>
> The sensitivity in Theorem 3.10 is the so-called _smooth_ sensitivity [1]. While the notion of smooth sensitivity relies, in turn, on that of local sensitivity, smooth sensitivity is not equivalent to local or global sensitivity. In fact, smooth sensitivity upper bounds local sensitivity by considering not only the point itself, but its neighborhood (see also Equation 1 in lines 287-288). This allows us to consider noise calibrated to the specific instance, the _local_ part of the definition, while providing formal DP guarantees that hold for the entire dataset.
>
> **Answer to Question 2:**
>
> $\mathbf{X}$ is the random variable that we sample from the pattern distribution.
>
> **Answer to Question 3:**
>
> We will explain the following for completeness; for expressivity the explanation is analogous. Lovasz' theorem states that if you have a vector with all the infinitely many homomorphism counts from all pattern graphs to the graph in question, this embedding is complete (i.e., it can be used to distinguish this graph from all other non-isomorphic graphs). As dealing with infinite vectors is not possible in practice, we sample patterns to obtain a finite-sized vector. This vector is expectation-complete, that is, in expectation we are able to distinguish as much as we could with the infinite-size vector. In more practical terms, this also means that the more patterns we sample, the better we get at distinguishing graphs.
>
> [1] Nissim et al. "Smooth sensitivity and sampling in private data analysis", 2007.
>
> Thanks again for your review and please let us know if our answers helped clarify any misunderstanding or if you have any additional questions about our contribution.

---

> > ### Comment · Reviewer_dNzD · 2025-11-26
> >
> > Thank you for the detailed response. The authors have adequately addressed my earlier concern, and I am satisfied with the clarification provided.
> >
> > I'm increasing my point to 8.

---

### Official Review · Reviewer_hgq5 · 2025-10-31

**Soundness:** 2
**Presentation:** 3
**Contribution:** 2
**Rating:** 4
**Confidence:** 4

**Summary:**

This paper investigates the theoretical trade-off between expressivity and privacy in graph representation learning. The authors propose constructing graph embeddings using noisy homomorphism densities, where homomorphism counts are perturbed with Gaussian noise calibrated via smooth sensitivity to ensure tCDP. Experiments are conducted on several OGB molecular datasets and synthetic graphs, showing that the proposed method maintains competitive predictive performance and effectively resists privacy attacks under reasonable privacy budgets.

**Strengths:**

1. Graph privacy is an important and practically relevant research direction.
2. The paper is relatively well-organized and logically coherent.

**Weaknesses:**

1. The paper emphasizes the use of homomorphism density vectors as a key tool. However, while it claims to fill the gap in understanding the interplay between expressivity and privacy in graph representation learning, it does not clearly define what this gap uniquely refers to. Prior studies (Sajadmanesh et al. 2021, 2023, 2024, as well as others) have already explored expressivity under differential privacy and aimed to design utility-preserving private graph learning frameworks.
2. The paper focuses on edge-level privacy and graph-level tasks without explaining why this specific setting was chosen. In practice, graph privacy leakage often involves node attributes, and for edge-level privacy, link prediction tasks are crucial for comprehensive evaluation. The current setup limits the practical relevance of the work.
3. The paper cites Lovász’s theorem (1967), which states that homomorphism counts can distinguish all non-isomorphic graphs. However, it later uses homomorphism densities, and in Remark B.1 admits that these cannot distinguish a graph from its blow-up. Although the authors propose adding “node counts” as a correction, they provide no theoretical proof that this modification preserves differential privacy guarantees or avoids leaking information about graph size. Moreover, the proposed fix only applies when graphs have different node counts—what about when they are the same?
4. Experimental details are incomplete. For instance, the paper sets $δ=10^{-6}$, $β=ρ'/5$, and $d=50$, but does not justify these choices or explain their sensitivity.
5. Experiments are limited to OGB molecular graphs and synthetic graphs, lacking evaluation on other types of graph data such as social networks (large, sparse) or medical graphs (sensitive node attributes). Since molecular graphs are relatively regular, the effectiveness of the proposed method on structurally complex graphs remains unverified, reducing generalizability.

**Questions:**

see weaknesses.

---

> ### Author Response · Authors · 2025-11-20
>
> We thank the reviewer for their time and their detailed feedback! We hope to address your concerns and clarify some possible misunderstandings on the scope and soundness of our submission in the following.
>
> **Addressing Weakness 1:**
>
> Thank you very much for the references. The mentioned papers, as you correctly point out, focus on developing methods to improve the trade-off between _utility_, measured with respect to some performance metric, and differential privacy. They do not, however, address the expressive power of the obtained embeddings. In the expressivity literature, the expressive power is quantified specifically as the capacity to distinguish non-isomorphic graphs, which is the definition we rely on; see, e.g., [1, Chapter 5] and [2]. In fact, Sajadmanesh et al. 2023 [3, point _(iv)_ in the conclusion] do advocate exploring the expressive power of private graph learning algorithms. While expressivity and utility are related, they are two distinct concepts and we do not strive to develop methods that beat SOTA algorithms in terms of utility, but rather to theoretically and empirically understand the _expressivity_-privacy trade-off. This is a research gap which is unexplored by the literature, and which we first investigate.
>
> **Addressing Weakness 2:**
>
> We study graph privacy primarily through the lens of _expressivity_. As mentioned in the previous point, expressivity analysis in graph representation learning generally focuses on quantifying the ability of an algorithm to distinguish structurally different graphs [1, Chapter 5]. Thus, we focus on graph-level tasks. Graphs that differ by a single edge have the _minimal_ possible structural difference which should be distinguished by an expressive algorithm. Therefore, we consider edge-level privacy as an ideal setting to investigate the underexplored interplay between privacy and expressivity.
>
> **Addressing Weakness 3:**
>
> Thank you for highlighting this additional contribution, which does not only apply to our own work, but also resolves a previous imprecision/mistake in Welke et al. [4]. First, note that our guarantees do not attempt to protect the size of graphs, but rather the presence/absence of individual edges. In fact, counting nodes in a graph is trivially differentially private with respect to edge privacy, as the node count is not affected by edges and is identical for two edge-neighboring graphs. If two graphs have the same number of nodes, we know that neither can be the blown-up version of the other, and the fix would not be necessary, as mentioned, e.g., in the proof of Theorem 4.1. We will include a more formal discussion in the main body of the paper in the revision.
>
> **Addressing Weakness 4:**
>
> In our submission, to the best of our knowledge, we provide all experimental details and code needed for reproducibility; please let us know if you noticed that some details are missing. A standard requirement is $\delta << 1/n$ [3] where $n$ is the number of edges in a graph in our setting. The choice of $\delta = 10^{-6}$ is commonly used in related literature and amply satisfies this. In fact, we have been quite conservative in this choice: we could increase the value of $\delta$, while still respecting $\delta << 1/n$, to obtain smaller values of $\epsilon$ for a fixed amount of noise (see Lemma A.2).  Our choice of $\delta$, together with the choice of $\beta$ and the range of values of $\rho’$, provide meaningful privacy guarantees and good performance for reasonable privacy budgets (see, e.g., $\epsilon = 1$ in  Fig. 2a in the main body and Fig. 3 in Appendix D.2). If you have specific alternative values in mind that could provide additional insights and strengthen our contribution, we would be happy to consider them. With respect to $d$, we consider 50 patterns following the empirical setup of [4]. We also explore additional values of $d$ in section D.3, and show that $d=50$ provides the best results across the tested values. We will make this discussion more explicit in the revision.

---

> ### Author Response · Authors · 2025-11-20
>
> **Addressing Weakness 5:**
>
> Thank you for bringing this up and allowing us to discuss our experiment design in more detail. We agree that the main focus of our paper is on our theoretical contribution: to the best of our knowledge, we are the first to establish a formal trade-off between expressivity and privacy in graph machine learning and, moreover, propose a first method on how to provide formal guarantees for both while retaining good predictive performance. We carefully designed our synthetic experiments to empirically verify our theoretical results on the trade-off between privacy and expressivity: for a fixed privacy budget, more expressive pattern classes require more noise and only help if more expressive power is required to solve the specific problem at hand. If you have additional suggestions on experiments that isolate this trade-off, we'd be happy to consider them. Approaches designed for large-scale graphs such as social networks or medical graphs generally focus on node classification, rather than graph classification, which is beyond the scope of our current work. Molecules are structurally diverse graphs that provide a natural setting where expressivity limitations can have concrete implications; e.g., consider the pairs of non-isomorphic molecules in Figure 3 and Figure 6 in [2] that are indistinguishable by standard GNNs. Thus, we chose to perform experiments on real-world molecule datasets, where we obtain good performance while ensuring both expressivity and privacy.
>
> [1] Pan and Leskovec, "The expressive power of graph neural networks", 2022.
> [2] Sato, "A survey on the expressive power of graph neural networks", 2020.
> [3] Sajadmanesh et al., "GAP: Differentially private graph neural networks with aggregation perturbation", 2023.
> [4] Welke et al., "Expectation-complete graph representations with homomorphisms", 2023.
>
> Thanks again for your detailed remarks, we hope we were able to address your concerns and are happy to engage in further discussion.

---

### Official Review · Reviewer_WKpZ · 2025-11-01

**Soundness:** 3
**Presentation:** 3
**Contribution:** 3
**Rating:** 6
**Confidence:** 3

**Summary:**

This paper investigates the trade-off between expressivity and privacy in graph representation learning. The authors propose using noisy homomorphism density vectors to achieve both distinguishability and differential privacy guarantees. The theoretical part derives upper bounds on sensitivity and smooth sensitivity, while the experimental section validates the privacy–utility trade-off on the OGBG and SBM datasets.

**Strengths:**

1. The paper demonstrates rigorous logical consistency from definitions and lemmas to theorem derivations, with solid mathematical details.
2. The proposed framework can be mapped to different GNN architectures (e.g., message-passing and subgraph-based GNNs) and allows for quantitative analysis of the trade-off.
3. The experiments show clear performance differences across pattern classes, validating the theoretical trends.

**Weaknesses:**

1. It only considers edge-level DP.
2. Experiments are conducted only on small-scale, toy-level datasets and do not demonstrate the feasibility of the approach on real-world large graphs or complex tasks.
3. The paper lacks comparison with established DP-GNN methods such as edge-level GAP, using only a randomly perturbed GNN as the baseline.
4. Computing homomorphism densities is computationally expensive. Please provide a detailed analysis of computational overhead.
5. The paper is clearly written, but it appears that the authors used last year’s ICLR template.

**Questions:**

See weakness above.


Since only edge-level privacy is considered, can this analysis be extended to node-level or subgraph-level differential privacy in the future?

---

> ### Author Response · Authors · 2025-11-20
>
> We thank the reviewer for their comments and their overall feedback that the soundness, presentation and contribution of our paper are **good**! Please find detailed answers to all your points below.
>
> **Addressing Weakness 1:**
>
> Yes, we guarantee edge-level privacy for graph representation learning. We argue that this is a common and important notion of graph privacy which is currently underexplored with respect to expressivity. Graphs that differ by a single edge have the _minimal_ possible structural difference which should be distinguished by an expressive algorithm, and are therefore an ideal setting to investigate edge-level DP. We leave an extension to node-level DP as possible future work.
>
> **Addressing Weakness 2:**
>
> We agree that the main focus of our paper is on our theoretical contribution: to the best of our knowledge, we are the first to establish a formal trade-off between expressivity and privacy in graph machine learning and, moreover, propose a promising method on how to provide formal guarantees for both while retaining good performance. However, the considered real-world datasets are not small-scale and toy-level: in fact, they are commonly used molecule benchmarks in expressivity analysis, which is the perspective of our investigation. Molecules provide a natural setting where expressivity limitations can have concrete implications; e.g., consider the pairs of non-isomorphic molecules that standard GNNs cannot distinguish in Figure 3 and Figure 6 in [1]. While we provide expressivity guarantees in expectation, private GNNs such as GAP provide no guarantees at all. Approaches designed for large-scale graphs such as social networks generally focus on node classification, rather than graph classification, which is beyond the scope of our current work.
>
> **Addressing Weakness 3:**
>
> Thank you for bringing up further baselines. Unfortunately, to the best of our understanding, GAP is a GNN architecture with unquantified expressive power and specifically designed for node classification on large scale graphs. Thus, GAP does not apply to our setting. Please let us know if you have suggestions for additional baselines which combine differential privacy and provable expressivity.
>
> In our work we adopt Randomized Response (RR) as a baseline, which is more appropriate for graph classification with edge privacy (cf. Hidano and Murakami [2]), to privatize the graphs and use them as input for  the provably expressive GIN [3]. In addition, we now also implemented and ran the recently introduced degree-preserving Randomized Response (DPRR) method [2] with GIN, which provides edge-level privacy by transforming the input graphs and is an improvement to the standard RR baseline. Please note that RR/DPRR are not directly comparable with our approach, as RR/DPRR do not rely on the smooth sensitivity framework we adopt and do not provide expressivity guarantees. Please find the results in the table below.
>
> | Method ($\epsilon = 1$) | $\texttt{MOLHIV}$ | $\texttt{MOLBBBP}$ | $\texttt{MOLBACE}$ | $\texttt{MOLLIPO}$ |
> |--|--|--|--|--|
> |  $\mathbf{t}(\mathbf{F}, G)$ (ours) | 0.692 $\pm$ 0.020  | 0.602 $\pm$ 0.005 | 0.652 $\pm$ 0.069  | 1.086 $\pm$ 0.004 |
> |  RR | 0.488 $\pm$ 0.008  | 0.440 $\pm$ 0.005 | 0.457 $\pm$ 0.024 | 1.568 $\pm$ 0.248 |
> |  DPRR | 0.595 $\pm$ 0.155 | 0.539 $\pm$ 0.019 | 0.648 $\pm$ 0.043 | 1.499 $\pm$ 0.333 |
>
> **Addressing Weakness 4:**
>
> Counting homomorphisms from arbitrary pattern graphs is #P-complete. However, if we restrict our patterns to graphs of bounded treewidth, which characterize the majority of expressive GNN architectures (cf. Table 1 in our submission), we can actually count them in polynomial time, parameterized by the treewidth [4]; please find a more detailed discussion in Appendix C.1 in our submission. We will elaborate on and move this discussion to the main part in the revised version. We will also provide runtimes for our computations.
>
> **Addressing Weakness 5:**
>
> We are glad you found the paper to be clearly written! Thanks for catching this, we have fixed the template in the revised version.
>
> **Questions**:
>
> _Since only edge-level privacy is considered, can this analysis be extended to node-level or subgraph-level differential privacy in the future?_
>
> Yes, this will be a great extension!
>
> [1] Sato, "A survey on the expressive power of graph neural networks", 2020.
> [2] Hidano and Murakami, "Degree-preserving randomized response for graph neural networks under local differential privacy", 2023.
> [3] Xu et al., "How Powerful are Graph Neural Networks?", 2019.
> [4] Díaz et al., "Counting H-colorings of partial k-trees", 2002.
>
> Thanks again for your review, please let us know if there is anything else we can address.

---

### Official Review · Reviewer_C4Rx · 2025-11-05

**Soundness:** 3
**Presentation:** 3
**Contribution:** 3
**Rating:** 4
**Confidence:** 4

**Summary:**

The paper investigates the trade-off between expressivity (ability to distinguish non-isomorphic graphs) and privacy (edge-level differential privacy) in graph representation learning. It proposes using homomorphism density vectors as embeddings, adding Gaussian noise calibrated via smooth sensitivity to achieve truncated concentrated DP (tCDP) guarantees. The authors argue that expectation-expressivity is preserved because the noise is mean-zero. Experiments on OGB molecular datasets and a synthetic SBM illustrate privacy–utility trade-offs and compare against a randomized-response baseline.

**Strengths:**

- The paper addresses an underexplored intersection of privacy and expressivity in graph learning.
- Formalizes expectation-expressivity and privacy guarantees using established DP frameworks.
- Demonstrates basic privacy–utility trade-offs and sanity-checks the insufficiency of global sensitivity.

**Weaknesses:**

- Experiments mostly use tree patterns; higher treewidth and complex pattern classes are barely explored. The empirical section lacks depth: pattern classes beyond treewidth 1 are barely tested, and runtime implications of homomorphism counting (#P-hard) are not addressed.
- No comparison to state-of-the-art DP-GNNs (e.g., ProGAP) or strong inference attacks like LinkTeller.
- Non-private use of degree bounds ($\Delta max$) for sensitivity calibration undermines DP guarantees. Furthermore, Δmax is estimated from data without privatization, which violates DP assumptions

**Questions:**

Given the strengths, I have the following concerns:

1. The authors claims that "Our embeddings match, in expectation, the expressive power of a broad range of graph neural networks (GNNs), such as
message-passing and subgraph GNNs, while providing formal privacy guarantees." but there is no comparison with GNN-based privacy methods like ProGAP or [1]. These are widely cited and provide edge-level DP guarantees. Can the authors include these baselines?

2. The paper evaluates privacy via nearest‑neighbor graph re‑identification on embeddings, ignoring stronger edge inference attacks. Why restrict to nearest-neighbor re-identification? Stronger attacks (e.g., LinkTeller) has shown non‑trivial leakage even against DP defenses for modest $\epsilon$ or modern membership/link inference analyses for GNNs; the empirical privacy validation thus seems under‑powered.

3. The sensitivity bound with degree dependence (Theorem 4.5) assumes a known max degree Δmax (e.g., fixed per dataset) or estimated (Chernoff) on synthetic data. Using estimated Δmax can itself leak information unless privatized or externally bounded, and the paper sets Δmax from data or heuristics in experiments, which is not privacy‑preserving if used to calibrate noise. This critical point is not addressed. Can the authors clarify this?

4. This is subtle but very important, does the method allow private release of embedding?

5. Most OGB experiments use tree patterns (tw=1), which undermines the empirical claim that “more expressive pattern classes” help under fixed privacy budgets. The brief MOLHIV experiment with small treewidth still tops out at tw=3; no cycles/graphlets on large graphs, no runtime/memory profiling of homomorphism computations, despite #P‑hardness beyond bounded treewidth.

6. Blow‑up invariance of densities is acknowledged in the appendix, but because densities cannot distinguish graph blow‑ups, the completeness claims require appending |V(G)| or switching to counts; this subtle but important caveat should be explicit in the main text where expressivity claims are made.



**Missing references:**

*On DP graph learning:*

[1]Olatunji et al. Releasing graph neural networks with differential privacy guarantees

[2]Xiang et al. Preserving Node-level Privacy in Graph Neural Networks

*On graph reconstruction attack:*

[3]Olatunji et al. Private graph extraction via feature explanations

[4]Zhou et al. On Strengthening and Defending Graph Reconstruction Attack with Markov Chain Approximation

---

> ### Author Response · Authors · 2025-11-20
>
> We thank the reviewer for their detailed comments and their overall feedback that the soundness, presentation and contribution of our paper are **good**! Please find detailed answers to all your questions below.
>
> **Answer to Question 1:**
>
> In the expressivity literature, the expressive power of a GNN (or, more generally, of a graph algorithm) is usually quantified as its ability to distinguish between non-isomorphic graphs. In our paper we _prove_ that our approach formally matches, in expectation, the expressive power of a wide class of GNNs studied in the expressivity literature. In contrast, existing work on private graph learning does not investigate expressivity. In fact, Sajadmanesh et al. 2023 [2, point _(iv)_ in the conclusion] does advocate for future work in this direction, i.e., to explore the expressive power of private graph learning algorithms.
>
> Thank you for suggesting ProGAP as an additional baseline. To the best of our knowledge, ProGAP is a GNN architecture with unquantified expressive power, specifically designed for node classification and link prediction, which is not our setting. PrivGNN is also designed for a node classification setting and relies, moreover, on access to unlabelled public data. Both baselines do not, therefore, align with our theoretical and empirical investigation. Moreover, both ProGAP and PrivGNN focus on privately releasing a GNN model, rather than a private embedding of the graphs themselves, while we produce private embeddings that can be used with any downstream algorithm. While an adaptation of ProGAP or PrivGNN to our graph classification setting and a quantification of their expressive power are surely interesting directions, we believe they are at the moment out of the scope of our investigation. This is why we decided to use Randomized Response (RR) as a baseline to privatize the graphs and then use GIN, a GNN with fully characterized expressive power, similarly to Hidano & Murakami [3]. If you believe we misunderstood the applicability of ProGAP or PrivGNN, please let us know how we could use it in our setting. We also welcome other suggestions for additional baselines which combine differential privacy and provable expressivity.
>
> As an additional baseline, we also implemented and ran the recently introduced degree-preserving Randomized Response (DPRR) [3] method with GIN: DPRR provides edge-level privacy by transforming the input graphs and is an improvement to the standard RR baseline. While RR/DPRR are still not directly comparable with our method, as they do not rely on smooth sensitivities and provide no expressivity guarantees, they are applied in a similar setting as ours.  Please find the results in the table below.
>
> | Method ($\epsilon = 1$) | $\texttt{MOLHIV}$ | $\texttt{MOLBBBP}$ | $\texttt{MOLBACE}$ | $\texttt{MOLLIPO}$ |
> |--|--|--|--|--|
> |  $\mathbf{t}(\mathbf{F}, G)$ (ours) | 0.692 $\pm$ 0.020  | 0.602 $\pm$ 0.005 | 0.652 $\pm$ 0.069  | 1.086 $\pm$ 0.004 |
> |  RR | 0.488 $\pm$ 0.008  | 0.440 $\pm$ 0.005 | 0.457 $\pm$ 0.024 | 1.568 $\pm$ 0.248 |
> |  DPRR | 0.595 $\pm$ 0.155 | 0.539 $\pm$ 0.019 | 0.648 $\pm$ 0.043 | 1.499 $\pm$ 0.333 |
>
> **Answer to Question 2:**
>
> We agree that evaluating against strong attacks is important; however, our goal is to isolate the expressivity-privacy trade-off in a model-agnostic way rather than to benchmark a specific GNN deployment scenario. Attacks like LinkTeller require an inference API on a trained GNN to determine pairwise influences between nodes.
> As we do not rely on a particular GNN to produce embeddings, nor expose a similar inference API, it is unclear to us how we could adapt LinkTeller or similar attacks to our setting; if you think there is a straightforward way to adapt LinkTeller to our setting, please let us know.
>
> In our attack scenario, we do not attack a model but rather we assume a quite strong attacker that has access to the original graphs so they can compute the noise free embeddings, _and_ has access to the noisy embeddings so they can try and match them to the original graphs. The attack we evaluate is a direct attack on the released DP embeddings, aligned with the neighboring-graph definition that we use in our theory.

---

> ### Author Response · Authors · 2025-11-20
>
> **Answer to Question 3:**
>
> Thank you for raising this subtle point. In our theoretical analysis, we rely on the fact that in many domains, there are _a priori_ constraints on the maximum degree of graphs which can be leveraged to obtain tighter bounds on the sensitivity of the homomorphism densities. Our theory therefore does not rely on a data-dependent, released estimate of the maximum degree but, rather, internally uses public information on the maximum degree to better choose the noise scale. For instance, molecules have to satisfy certain chemical valence constraints, and therefore can only have a specific max degree which is specific to the class of graphs admitted by certain molecule classes. We would like to thank you again for raising this point as it made us notice that, in practice, the upper bound derived with the max degree can even be too conservative: if for a specific graph $\Delta_{\max}>n$, then setting $\Delta_{\max}=n$ in Equation (3) leads to a tighter bound on the sensitivity. This does not affect the general discussion or the differential privacy of our embeddings, but could lead to slightly better empirical results in some cases.
>
> For the synthetic dataset, our bound on the max degree is obtained from the SBM parameters and not from the graphs themselves.
>
> For domains where no meaningful public degree bound is available, one could still estimate a usable $\Delta_{\max}​$ privately or, simply, put $\Delta_{\max}=n$ in Equation (3). For a private estimate, one may add, e.g., Laplacian noise to the empirical maximum degree under a small additional privacy budget.
>
> We will make this discussion more explicit in our revision.
>
> **Answer to Question 4:**
>
> Yes, the embeddings can absolutely be released; we will highlight this crucial theoretical contribution in our revision. In contrast to many private GNNs, where randomization/privatization happens during the forward pass of the GNN, we prove that the noisy homomorphism density _embeddings_ are _private_ (and _expressive_), and can therefore be released. This means, you can perform any downstream task/analysis with our embeddings and retain differential privacy thanks to the post-processing property. Thus, while the initial computation of the embeddings can be costly, for us this is a one-time step, while for most private GNNs the privatization happens during training and has to be always recomputed if, e.g., the same data has to be used for a different task.
>
> **Answer to Question 5:**
>
> Thank you for bringing this up and allowing us to discuss our experiment design in more detail. In our work, we **do not** claim that more expressive pattern classes generally help under fixed privacy budgets (see also point _(ii)_ in the paragraph _Main contributions_). In fact, we prove that more expressive pattern classes require more noise for a fixed privacy budget. Therefore, for a fixed privacy budget, we expect patterns from more expressive graph classes to _only_ help if more expressive power is required to solve the specific problem at hand. This is precisely why we carefully designed the experiments on SBM data where, provably, cycles are sufficient to distinguish the SBM graphs, while trees, provably, are not. If you have additional suggestions on experiments that isolate this trade-off, we'd be happy to consider them.
>
> With respect to the pattern classes we investigate: cycles, for instance, have treewidth 2 so they are a subset of the class of all graphs with treewidth 2. Treewidth 3 graphs already include a large variety of graphs and match the expressive power of most (tractable) expressive GNNs [4]. Even in practical applications for large-scale graphs, which are beyond the scope of our investigation, the considered graphlets usually do not exceed treewidth 3 [5] [6, Figure 4]. It is true that for large graphs, counting homomorphism for very expressive patterns becomes unfeasible as it scales as $\mathcal{O}(n^{t+1})$, where $t$ is the treewidth of the pattern and $n$ denotes the number of nodes. Please be aware that this is not unique to our method: Increased expressive power generally comes at the cost of increased time complexity (yet another interesting trade-off!). For instance, the time complexity of $k$-GNNs, which match the expressive power of $k$-WL (the most widely used algorithm for graph isomorphism testing to date) scales as $\mathcal{O}(n^{k+1})$.
>
> To summarize, we specifically chose bounded treewidth patterns as they characterize the expressive power of most (expressive) GNN architectures. Moreover, we want to stress the fact that the counting step only needs to be performed once, after which the resulting embeddings can be used for any downstream tasks. Sections 2.5 & 4.4 in Nguyen and Maehara [7] and Section 4 & Appendix C in Welke et al. [8] provide more thorough analyses on the runtime. We will nevertheless provide runtimes for our computation.

---

> ### Author Response · Authors · 2025-11-20
>
> **Answer to Question 6:**
>
> Thanks for pointing this out. We will include a more formal discussion in the main body of the paper. We would also like to point out that this observation and our fix also resolves a previous imprecision/mistake in Welke et al. [4] on the expressive power of homomorphism densities.
>
> [1] Olatunji et al., "Releasing graph neural networks with differential privacy guarantees", 2021.
> [2] Sajadmanesh et al., "GAP: Differentially private graph neural networks with aggregation perturbation", 2023.
> [3] Hidano and Murakami, "Degree-preserving randomized response for graph neural networks under local differential privacy", 2023.
> [4] Zhang et al. "Beyond Weisfeiler-Lehman: A Quantitative Framework for GNN Expressiveness", 2024.
> [5] Maehara and Hoang, "Deep homomorphism networks", 2024.
> [6] Yanardag and Vishwanathan, "Deep graph kernels", 2015.
> [7] Nguyen and Maehara, "Graph homomorphism convolution", 2020.
> [8] Welke et al., "Expectation-complete graph representations with homomorphisms", 2023.
>
> Thank you very much again for the pointers to additional literature, we will incorporate the papers in a revised version of our submission. Please let us know if any doubt or question was left unaddressed, we are happy to engage in further discussion if anything is unclear.

---

### Author Response · Authors · 2025-11-14

We want to thank all reviewers for their time invested in the review process and appreciate their comments and positive feedback!

We are currently working on carefully addressing all responses and will provide comprehensive answers as soon as possible.

---

### Author Response · Authors · 2025-11-18
**Global reply to reviews**

Thanks again for your reviews!

In this global reply, we aim to address common points raised by the reviewers and, in particular, clarify the role of _expressivity_ in our analysis. We will also reply to each reviewer individually.

* **Expressivity and its connection to privacy in graph representation learning:** We specifically study the trade-off between graph privacy and _expressivity_. Although expressivity and privacy have been independently studied extensively for graph learning algorithms, their interplay has not been formally investigated so far. Indeed, Sajadmanesh et al.  [1, point _(iv)_ in the Conclusion] highlighted that there is a lack of research on the expressive power of differentially private graph learning algorithms.
* **Choice of problem setting:** As expressivity analysis focuses on the ability to distinguish between structurally different, i.e., non-isomorphic graphs, we focus on graph-level tasks. Graphs that differ by a single edge have the _minimal_ possible structural difference which should be distinguished by an expressive algorithm. Therefore, we consider _edge_-privacy as an ideal setting to investigate the underexplored interplay between privacy and expressivity.
* **Choice of datasets:** We focused our experiments on molecular graph classification benchmark datasets, because they are standard in the expressivity literature, which is the perspective of our theoretical investigation. Molecules provide a natural setting where expressivity limitations can have concrete implications (e.g., non-isomorphic molecules that standard GNNs cannot distinguish). That said, as suggested by the reviewers, we are currently running additional experiments on non-molecular data and will report detailed results as soon as they are available.
* **Choice of baselines:** We thank the reviewers for the additional GNN baseline suggestions. Unfortunately, the proposed baselines (GAP [1], ProGAP [2] and PrivGNN [3]) are GNN architectures with unquantified expressive power and specifically designed for node classification on large scale graphs, which is not our setting. In our work, we adopt a randomized response (RR) baseline appropriate for graph classification with edge privacy (cf. Hidano and Murakami [4]), to privatize the graphs and feed them into the provably expressive Graph Isomorphism Network [5]. Following the reviewers' suggestions for additional baselines, we implemented and ran the recently proposed degree-preserving randomized response (DPRR) method [4]. We report the results in the table below. DPRR improves upon randomized response, but is still worse than our method - which in addition to differentially privacy guarantees also ensures expressivity in expectation.
| Method ($\epsilon = 1$) | $\texttt{MOLHIV}$ | $\texttt{MOLBBBP}$ | $\texttt{MOLBACE}$ | $\texttt{MOLLIPO}$ |
|--|--|--|--|--|
|  $\mathbf{t}(\mathbf{F}, G)$ (ours) | 0.692 $\pm$ 0.020  | 0.602 $\pm$ 0.005 | 0.652 $\pm$ 0.069  | 1.086 $\pm$ 0.004 |
|  RR | 0.488 $\pm$ 0.008  | 0.440 $\pm$ 0.005 | 0.457 $\pm$ 0.024 | 1.568 $\pm$ 0.248 |
|  DPRR | 0.595 $\pm$ 0.155 | 0.539 $\pm$ 0.019 | 0.648 $\pm$ 0.043 | 1.499 $\pm$ 0.333 |

* **Degree bound:** In our experiments, we take advantage of the fact that, in many domains such as that of molecular graphs, there are _a priori_, public constraints on the maximum degree of graphs, which can be leveraged to obtain tighter bounds on the sensitivity of the homomorphism densities. We want to point out that our theoretical results do not depend on this. In any case, if needed, one may privately estimate the max degree by adding, e.g., Laplacian noise, to the empirical maximum degree under a small additional privacy budget.
* **Time complexity of homomorphism counting:** For general graphs, counting homomorphisms is #P-complete. However, for many pattern classes (e.g., paths, stars, cycles) they can be computed efficiently. We specifically chose bounded treewidth graphs as they already characterize the expressive power of most GNN architectures. Moreover, we want to stress the fact that the counting step only needs to be performed once, after which the resulting embeddings can be used for any downstream tasks. Sections 2.5 & 4.4 in Nguyen and Maehara [6] and Section 4 & Appendix C in Welke et al. [7] provide more thorough analyses on the runtime. We will nevertheless provide runtimes for our computation.

---

> ### Author Response · Authors · 2025-11-18
>
> **References:**
>
> [1] Sajadmanesh et al., "GAP: Differentially private graph neural networks with aggregation perturbation", 2023.
>
> [2] Sajadmanesh et al., "Progap: Progressive graph neural networks with differential privacy guarantees", 2024.
>
> [3] Olatunji et al., "Releasing graph neural networks with differential privacy guarantees", 2021.
>
> [4] Hidano and Murakami, "Degree-preserving randomized response for graph neural networks under local differential privacy", 2023.
>
> [5] Xu et al., "How Powerful are Graph Neural Networks?" 2019.
>
> [6] Nguyen and Maehara, "Graph homomorphism convolution", 2020.
>
> [7] Welke et al., "Expectation-complete graph representations with homomorphisms", 2023.

---

### Author Response · Authors · 2025-11-25
**Additional experiments and revised version**

**Additional experiments**

Following the reviewers' suggestions, we conducted additional experiments on the three network datasets $\texttt{REDDIT-BINARY}$, $\texttt{REDDIT-MULTI-5K}$ and GitHub $\texttt{STARGAZERS}$. We chose these datasets based on the related work closest to our setting, Hidano and Murakami [1], who also consider graph-level tasks with edge-level privacy. We computed homomorphism density embeddings from pattern graphs with treewidth $1$, which matches the expressive power of standard message-passing graph neural networks. Note that, in contrast to our method, the results in [1] do not provide any formal guarantee on the expressivity of the embeddings. All experiments were run on a single consumer-grade machine for a total of 9 runs. In the table below, we present results for $\epsilon=1$ and $\delta=10^{-6}$.

| | $\texttt{REDDIT-BINARY}$ | $\texttt{REDDIT-MULTI-5K}$ | $\texttt{STARGAZERS}$ |
|--|--|--|--|
| **Accuracy** | 0.758 $\pm$ <0.001  | 0.416 $\pm$ 0.021 | 0.590 $\pm$ 0.015 |
| **AUC** | 0.775 $\pm$ 0.009  | 0.749 $\pm$ 0.014 | 0.609 $\pm$ 0.026 |

We perform similarly to [1, Figure 5], who achieve an accuracy of $\approx 72$, $\approx 41$, and $\approx 61$ on $\texttt{REDDIT-BINARY}$, $\texttt{REDDIT-MULTI-5K}$ and GitHub $\texttt{STARGAZERS}$, respectively. Note that we rely on simple classifiers ($k$-NN and Random Forest), which further confirms that our private homomorphism density embeddings are informative and useful in practice. Once the embeddings have been obtained, the training procedure can therefore be very fast. In contrast, the approach in [1] can be computationally demanding, and we could not run their method on our machine due to out-of-memory errors. This is not surprising as these datasets are significantly larger than the $\texttt{OGBG}$ ones, with $\texttt{REDDIT-BINARY}$, $\texttt{REDDIT-MULTI-5K}$ and GitHub $\texttt{STARGAZERS}$ having an average number of $\approx 430$, $\approx 509$, and $\approx 114$ nodes per graph, respectively. Our additional results thus show that homomorphism densities can be used on larger graphs outside of the molecular domain, and thus showcase the promising practical implications of our theoretical results.

**Runtime**

We measured the time it takes to compute homomorphism density embeddings for $\texttt{MOLHIV}$ with increasing maximum treewidth of {$1, 2, 3$} and $\texttt{REDDIT-BINARY}$ with treewidth $1$. The results are presented in Table 8 of the revised paper: for our hardware setup, homomorphism counting is comparable to running GIN with RR/DPRR for roughly 200 epochs. Since the embeddings we obtain are highly informative, as well as provably private and expressive, the training time is then negligible (in the order of seconds) as we can rely on simple classifiers such as $k$-NN: our theoretical contribution therefore also shows competitive practical utility.

**Revised version**

We also uploaded a revised version of our paper that addresses the questions raised by the reviewers, as well as includes the additional experiments and runtime comparisons. We highlighted the edits to make them easy to identify.

We want to thank the reviewers again for their suggestions, which we think substantially helped us strengthen our contribution.

[1] Hidano and Murakami, "Degree-preserving randomized response for graph neural networks under local differential privacy", 2024.

---

### Author Response · Authors · 2025-11-30
**New rebuttal process and AC assignment**

Dear AC,

first of all: thank you for taking on this new role to ensure a fair outcome of the rebuttal. This has been a frustrating process for authors and reviewers alike, and we're sure that also extends to ACs.

As we are also a bit unsure of the process now, we decided to ask you directly about the next steps. We plan to post a summary of our rebuttal + reviewer answers at the end of the rebuttal period. We believe we have addressed all concerns raised by the reviewers but unfortunately only one reviewer had the chance to answer before the freeze, so please do let us know if you think there is anything left unaddressed or that would benefit from more discussion.

Let us know if there is anything we can do to make the process easier, and thank you again for stepping in!

---

### Author Response · Authors · 2025-12-03
**Summary of discussion period**

As the discussion period draws to an end, we'd like to summarize our contributions, the reviews, and how we addressed open issues:

* We formally analyze the interplay between _expressivity_ and _privacy_. These notions have been independently studied extensively in graph machine learning, but we are the first to characterize their trade-off: for a fixed privacy budget, more expressive pattern classes require more noise and only help if more expressive power is required to solve the specific problem at hand.
* We provide private and expressive-in-expectation graph embeddings by building on the concept of graph homomorphism counts and results from extremal graph theory. In addition to our theoretical guarantees, our embeddings also have _great practical utility_, as showcased by our experiments on molecular as well as network datasets.
* The embeddings we propose can be _privately released_ and therefore used for any downstream analysis thanks to the post-processing property of differential privacy. This is an improvement over many private graph machine learning algorithms, where the privatization takes place during the forward pass/training and has to be computed anew for further passes.

We are happy that reviewers judged our soundness, presentation, and contribution to be **good** (**C4Rx** and **WKpZ**), and remarked that we _"provide a strong theoretical guarantee for the expressivity and privacy of graph representation learning"_ (**dNzD**). Reviewers emphasized that we investigate the _"underexplored intersection"_ (**C4Rx**) of expressivity and privacy, _"an important and practically relevant research direction"_ (**hgq5**), and praised our experiments for clearly _"validating"_ our theoretical investigation (**WKpZ**) and for highlighting the _"validity and soundness of the theoretical claims"_ (**dNzD**).

As the rebuttal phase was frozen early, reviewers **C4Rx**, **WKpZ** and **hgq5** did not have a chance to answer our rebuttal. We next summarize how we believe we exhaustively addressed all their concerns.

* We performed experiments on three additional network datasets with larger graphs (to address **WKpZ** and **hgq5**). **Our method is competitive with recent work that considers a problem setting compatible with ours** (see Table 4 in the revision) **and, in addition, provides expressivity guarantees**.
* We performed additional experiments with the DPRR baseline (to address **C4Rx** and **WKpZ**). **DPRR improves upon the original RR baseline, but still performs worse than our method** (see Table 2 in the revision).
* We added a more in-depth discussion of the time complexity to the main body, and measured the time it takes to compute the homomorphism densities (to address **C4Rx** and **WKpZ**). **On, e.g., $\texttt{MOLHIV}$, the runtime of our method is comparable to training a standard GNN for 200 epochs** (see Table 8 in the revision).
* We made the motivation for our problem setting clearer (to address **WKpZ** and **hgq5**). We consider **graph-level tasks with edge-level privacy since graphs that differ by a single edge have the minimal possible structural difference which should be distinguished by an expressive algorithm**.
* We added a more detailed discussion on blowup graphs to the main body of the paper (to address **C4Rx** and **hgq5**), where we highlight that **we can distinguish between graphs and their blowups by simply appending node counts to our homomorphism density embeddings while remaining edge private**.

---

### Meta-Review · Area_Chair_aYEh · 2026-01-07

**Summary:**

This paper formally investigates the trade-off between edge-level differential privacy and graph expressivity using homomorphism density vectors. It theoretically proves that these embeddings preserve expressivity in expectation under privacy noise calibrated via smooth sensitivity, supported by experiments on molecular and social network datasets.

**Reviewer Concerns:**

The rebuttal successfully addressed concerns regarding the validity of the privacy guarantee (smooth vs. local sensitivity), the use of degree bounds (public constraints vs. data-dependent), and the lack of non-molecular benchmarks. The primary outstanding friction is the preference of some reviewers for empirical comparisons against heuristic DP-GNNs, though the authors justify their focus on provable expressivity.

**Reviewer Scores:**

dNzD: Explicitly commented that they were satisfied with the clarification on smooth sensitivity and intended to raise their score to 8.

C4Rx: Would likely raise their score to 5 or 6, as the authors clarified that $\Delta_{max}$ relies on public domain constraints (negating the privacy leak concern) and added the requested DPRR baseline.

WKpZ: Would likely maintain a 6 or rise to 7, as the additional experiments on larger datasets (IMDB/GitHub) and runtime analysis addressed their concerns about scalability and scope.

hgq5: Would likely increase to 5 or 6 after the authors clarified the distinct theoretical contribution (expressivity-privacy trade-off vs. utility-privacy) and addressed the graph blow-up indistinguishability issue.

---

### Decision · Program_Chairs · 2026-01-26

Accept (Poster)